# FaSS-MVS: Fast Multi-View Stereo with Surface-Aware Semi-Global Matching from UAV-Borne Monocular Imagery

**DOI:** 10.3390/s24196397

**Published:** 2024-10-02

**Authors:** Boitumelo Ruf, Martin Weinmann, Stefan Hinz

**Affiliations:** 1Fraunhofer Institute of Optronics, System Technologies and Image Exploitation IOSB, 76131 Karlsruhe, Germany; 2Institute of Photogrammetry and Remote Sensing, Karlsruhe Institute of Technology, 76131 Karlsruhe, Germany; martin.weinmann@kit.edu (M.W.); stefan.hinz@kit.edu (S.H.)

**Keywords:** multi-view stereo, plane-sweep multi-image matching, semi-global optimization, surface-awareness, online processing, oblique aerial imagery, UAVs

## Abstract

With FaSS-MVS, we present a fast, surface-aware semi-global optimization approach for
multi-view stereo that allows for rapid depth and normal map estimation from monocular aerial
video data captured by unmanned aerial vehicles (UAVs). The data estimated by FaSS-MVS, in turn, facilitate online 3D mapping, meaning that a 3D map of the scene is immediately and incrementally generated as the image data are acquired or being received. FaSS-MVS is composed of a hierarchical processing scheme in which depth and normal data, as well as corresponding confidence scores, are estimated in a coarse-to-fine manner, allowing efficient processing of large scene depths, such as those inherent in oblique images acquired by UAVs flying at low altitudes. The actual depth estimation uses a plane-sweep algorithm for dense multi-image matching to produce depth hypotheses from which the actual depth map is extracted by means of a surface-aware semi-global optimization, reducing the fronto-parallel bias of Semi-Global Matching (SGM). Given the estimated depth map, the pixel-wise surface normal information is then computed by reprojecting the depth map into a point cloud and computing the normal vectors within a confined local neighborhood. In a thorough quantitative and ablative study, we show that the accuracy of the 3D information computed by FaSS-MVS is close to that of state-of-the-art offline multi-view stereo approaches, with the error not even an order of magnitude higher than that of COLMAP. At the same time, however, the average runtime of FaSS-MVS for estimating a single depth and normal map is less than 14% of that of COLMAP, allowing us to perform online and incremental processing of full HD images at 1–2 Hz.

## 1. Introduction

The image-based estimation of depth maps and geometry by dense image matching (DIM) and multi-view stereo (MVS) is one of the fundamental tasks in photogrammetry, remote sensing and computer vision. It enables a wide range of high-level applications such as navigation and path planning for autonomous mobile robot (AMR) systems, urban planning and monitoring, simulation and 3D modeling, as well as virtual, mixed and augmented reality. The ongoing development and increasing availability of commercial off-the-shelf (COTS) UAVs is opening up new possibilities and applications for image-based 3D mapping, both offline and online. In recent years, for example, the use of COTS UAVs by emergency services such as firefighters and medical rescue services has been steadily increasing, which in turn facilitates rapid and large-scale situation assessment or enables monitoring of areas inaccessible to ground forces [1,2]. In this context, image-based techniques and photogrammetry based on aerial reconnaissance are a key element in assisting the rescue workers, provided that the environmental conditions, e.g., weather and daytime, allow a visual inspection [2].

Meanwhile, there is a large collection of software toolboxes, such as COLMAP [3,4] or OpenMVS (http://cdcseacave.github.io, accessed on 27 August 2024), for performing offline photogrammetric 3D reconstruction, allowing to accurately reconstruct the disaster site from aerial imagery. Highly accurate 3D reconstructions can be used to accurately assess the damage caused by an incident or the structural integrity of a partially collapsed building. First responders, however, require rapid and extensive 3D mapping of the disaster site in a short period of time, rather than a highly accurate 3D reconstruction. This allows them to quickly assess the situation, prioritize resources, and plan routes and operations through hard-to-reach areas.

MVS methods can be divided into three categories based on the resulting scene representation, namely volumetric, point cloud-based, and depth map-based [5]. While the first two categories typically work on the full extent of the scene, depth map-based methods typically separate the depth estimation process from the depth map fusion process. This makes such methods typically more versatile, especially with respect to online and iterative 3D mapping, since the depth map estimation can be performed on a locally limited set of images, resulting in separate depth maps that can be subsequently fused into different scene representations, e.g., true orthophoto and 2.5D height map [6], 3D point cloud or mesh [7]. With this in mind, this work proposes and investigates an approach for fast multi-view stereo, by combining the SGM algorithm with a true multi-image matching approach. In it, we propose to:use an efficient plane-sweep sampling to perform hierarchical dense multi-image matching;adopt the SGM algorithm to work with depth hypotheses generated by plane-sweep sampling;extend the SGM algorithm to favor not only fronto-parallel surfaces in the computation of dense depth maps, by incorporating a surface-aware regularization based on local surface normals;implement and deploy it on modern GPU hardware to efficiently compute dense depth, normal and confidence maps online from image sequences.

Our fast, surface-aware semi-global optimization approach for multi-view stereo (FaSS-MVS) is designed to assist specialized first responders in deploying a high-end COTS UAV in combination with a ground control station (GCS) to rapidly assess the situation through aerial reconnaissance. It is assumed that the image data are streamed down to the GCS during the operation of the UAV, where they can be processed by more powerful hardware. Even though this approach is proposed with the above use case in mind, it is not limited to airborne data and can also be used to perform incremental and online 3D mapping by a ground-based robot or sensor system. Although learning-based approaches using deep neural networks [5,8,9,10] have made significant improvements in recent years, with FaSS-MVS we still rely on a traditional processing pipeline to ensure a high reliability and explainability in a practical application. We evaluate FaSS-MVS on two public datasets for dense MVS with accurate ground truth, and on two use-case-specific datasets. It combines and extends our previous work presented in Ruf et al. [11,12], by:a more detailed description of the algorithms used;extending the plane-sweep multi-image matching to use non-fronto-parallel plane orientations;improving the surface-aware regularization of the SGM algorithm;using a different confidence measure for estimating the confidence map;a thorough evaluation and ablation study with respect to different aspects and configurations of the approach;providing a detailed discussion with respect to the support of rescue workers by aerial reconnaissance.

An earlier and in some parts more elaborate version of this work has already been published as part of the PhD thesis by Ruf [13]. In contrast to the first publication, we have extended the evaluation and comparison of FaSS-MVS with respect to related approaches from the literature.

### 1.1. Paper Outline

This paper is organized as follows. In Section 1.2, the related work on incremental image-based 3D mapping for online processing as well as the reconstruction of non-fronto-parallel surfaces using DIM and MVS are briefly summarized. In this, it is also delineated how the presented approach differs from those presented in the related work. In Section 2, the entire processing pipeline of the presented approach is illustrated and outlined with a short overview. This is followed by a detailed description of the implementation and methodology of each step of the processing pipeline as well as a description of the datasets and error metrics used for evaluation. The results of the conducted experiments are presented in Section 3. Subsequently, the results are discussed in Section 4 and put into context of the considered use case, before a summary and concluding remarks, as well as a short outlook on future work, are given in Section 5.

### 1.2. Related Work

Due to the ever-increasing demand for detailed 3D models, the research in the fields of photogrammetry, remote sensing and computer vision has brought up a number of software suites and applications that focus on estimating accurate and dense depth and geometry information from a large set of input images using DIM and MVS. Prominent and widely used representatives of such applications are MVE [14], PMVS [15], SURE [16,17], COLMAP [4], ACMMP [18], and OpenMVS, to name a few. However, these approaches are designed for offline processing, with the goal of accuracy and completeness of the resulting 3D model, assuming that all input data are available at the time of reconstruction and that there are no critical constraints on computation time or hardware resources.

In contrast, the goal of FaSS-MVS is to extract dense depth and geometry information from image sequences as they are acquired, or at least while the image data stream is being received, in case direct processing is not possible due to the acquisition by a small UAV and its limited hardware resources. The focus is therefore on incremental and online processing of the input data by DIM and MVS.

#### 1.2.1. Incremental Camera-Based Mapping for Online Processing

Early work on incremental and online camera-based mapping of the local environment was primarily by robotics and augmented reality (AR) applications [19,20,21]. The main goal was to robustly localize the camera pose, and thus the sensor carrier, with respect to its environment in order to navigate through the environment or to augment the camera images with additional information. Since the focus of these so-called simultaneous localization and mapping (SLAM) algorithms is on estimating the camera pose and trajectory, the detailed and dense mapping of the environment was rather of secondary interest. In turn, these approaches relied mainly on point features for tracking and mapping rather than direct pixel matching. Since a dense and detailed model of the environment is essential for a convincing AR experience, subsequent work [22,23] has proposed dense mapping simultaneously with image acquisition and camera localization. However, these approaches aim at reconstructing rather small-scale environments and thus use short baseline video clips for image matching, which in turn allows relying on dense optical flow methods to find dense pixel correspondences [22]. In contrast, the input to the approach presented in this paper is assumed to be image data captured by a UAV, typically flying several tens of meters away from the object of interest. The approach presented here is thus designed to densely map a large-scale environment, which in turn requires image matching on a wide baseline, rather than tracking pixel-by-pixel correspondences between successive frames.

The works of Gallup et al. [24] and Pollefeys et al. [25] are part of the early approaches to camera-based mapping and reconstruction of urban environments. They used the plane-sweep algorithm [26] for true multi-image matching to map and reconstruct building facades in real time from images captured by a vehicle-mounted camera. They rely on vanishing points detected in the input images and data from an additional inertial measurement unit (IMU) to recover the orientations of the building facades and the ground plane relative to the camera. To find the optimal plane configuration for each pixel and, in turn, extract a depth map from the results of the DIM, Pollefeys et al. [25] employ a Bayesian formulation with a subsequent selection of the winner-takes-it-all (WTA) solution, while Gallup et al. [24] minimize a formulated energy functional. Other approaches to urban reconstruction from ground-based imagery, such as those of Furukawa et al. [27], Sinha et al. [28], and Gallup et al. [29], perform piecewise planar reconstruction by fitting multiple differently oriented planes into the scene and optimizing photometric consistency. They minimize an energy functional using a graph-cut algorithm that takes a few minutes on a commodity CPU.

More recent approaches to online camera-based 3D mapping are presented by Kern et al. [6] and Zhao et al. [30]. In their work, the authors propose complete processing pipelines for online and real-time 3D mapping from aerial imagery, which are highly relevant for the use case outlined in this paper. The design of the processing pipelines is similar to the approach of Pollefeys et al. [25], consisting of camera pose estimation, DIM and depth map fusion. For the image-based DIM, Kern et al. [6] relies on the so-called PlaneSweepLib [31], which is based on the work of Gallup et al. [24] and Pollefeys et al. [25]. In contrast, the focus of RTSfM [30] is on efficient and globally consistent Structure-from-Motion (SfM) in real time. For the task of DIM, RTSfM relies on the two-view stereo approach ELAS [32], which is run on image pairs to estimate the depth maps.

The presented approach also uses the plane-sweep algorithm to perform efficient dense multi-image matching. The use of a plane-sweep algorithm for the task of DIM is mainly motivated by its ability to generate depth hypotheses by matching an arbitrary number of input images, as well as the fact that it can be efficiently optimized for massively parallel execution on GPUs, making it particularly suitable for online processing. The use of COTS UAVs as a sensor carrier introduces the need to efficiently handle large scene depths and thus potentially large sampling spaces, due to the relatively low flight altitude and the ability to freely pitch the camera. To limit the sampling space and thus the number of depth hypotheses generated, we embed the plane-sweep algorithm in a hierarchical processing scheme.

#### 1.2.2. Efficient Dense Image Matching Accounting for Non-Fronto-Parallel Surfaces

The so-called SGM algorithm proposed by Hirschmüller [33,34] has become one of the most widely used approaches for both online and offline DIM due to its efficiency and convincing results [16,17,35,36,37]. In their work, Sinha et al. [38] combine plane-sweep multi-image matching with the SGM algorithm to estimate dense and highly accurate disparity maps. In contrast to the presented approach, Sinha et al. [38] use local slanted planes extracted from feature correspondences to generate disparity hypotheses and use the SGM algorithm to recover a disparity map. They evaluate their approach on a high-resolution stereo benchmark and achieve a significant improvement over the standard SGM algorithm in terms of both runtime and accuracy. The runtime improvement is attributed to the fact that the local plane-sweep allows us to test a locally limited part of the full disparity range for each pixel, thus reducing the computational complexity of the optimization within the SGM algorithm. Similar improvements to overcome the problem of high computational complexity due to the large disparity range inherent in oblique aerial imagery were made by Haala et al. [37] by embedding the SGM in a hierarchical coarse-to-fine processing.

Although many urban environments can be well abstracted by piecewise planar reconstructions, not all structures are fronto-parallel, i.e., their surface orientations are not parallel to the image plane. The original formulation of the SGM algorithm, however, only models a first-order smoothness term and thus favors fronto-parallel surfaces, leading to staircase artifacts when reconstructing slanted surfaces. This should be avoided, especially if the goal is a visually appealing reconstruction of the environment. While Hermann et al. [39] and Ni et al. [40] propose to include a second-order smoothness assumption in the formulation of the SGM energy function, Scharstein et al. [41] proposes a simpler yet effective improvement to address this issue. Specifically, plane priors, which can be recovered from normal maps or point correspondences, are used to adjust the zero-cost transition within the path aggregation of the SGM, thus penalizing deviations from the surface orientation represented by the prior. The major advantage over the other approaches is that the pixel-wise offset for the zero-cost transition can be computed in advance.

In this work, we use an improved implementation of the SGM algorithm to regularize the cost volume and efficiently extract an accurate dense depth map from the pixel-wise depth hypotheses generated by the plane-sweep DIM. We also adopt the approach presented by Scharstein et al. [41] to account for non-fronto-parallel surfaces by adjusting the zero-cost transition based on surface information stored in a normal map. Moreover, we also propose to reduce the fronto-parallel bias of the SGM algorithm by adjusting the zero-cost transition in the path aggregation based on the gradient of the minimum-cost path. And just like Haala et al. [37], we also embed the SGM in a hierarchical coarse-to-fine processing. Very similar to FaSS-MVS seems to be the approach of Roth and Mayer [42]. They also rely on the improvements proposed by Scharstein et al. [41] and combine the SGM with a plane-sweep DIM. However, their work focuses on estimating disparity images from ground-based stereo image pairs and has only been evaluated on synthetic scenes.

#### 1.2.3. Learning of Dense Image Matching and Multi-View Stereo Reconstruction

Due to the success of deep-learning-based methods in other areas of computer vision and photogrammetry, the technological advances gained have also been transferred and applied to the task of DIM and MVS, resulting in approaches [5,8,9,43] that outperform state-of-the-art model-based approaches on numerous common benchmarks. Despite recent improvements and highly accurate results, all of these approaches are trained in a supervised manner and thus require datasets with appropriate ground truth. However, the availability and versatility of appropriate datasets is not very high, especially with respect to real-world scenarios, which still greatly hinders the practical use of deep-learning-based MVS approaches. To overcome this problem, recent approaches [10,44] attempt to train models in an unsupervised, or sometimes referred to as self-supervised, manner. But again, their practical use and ability for generalization still needs more studies [44]. These limitations are the reasons why learning-based approaches for the task of MVS are not yet practical for the considered use case, namely to reliably support emergency forces in incremental and online mapping of the operational area. In addition, we believe that there are still a number of aspects related to traditional MVS approaches that need to be addressed, such as runtime or fronto-parallel bias, which we aim to address in this work.

## 2. Materials and Methods

The processing pipeline of FaSS-MVS is outlined in Figure 1. Given an input bundle ${\left(I,P\right)}_{k}\in \Omega$, consisting of *k* input images I extracted in sequential order from an image sequence, and corresponding camera poses P, our approach computes depth, normal, and confidence maps $\left(D,N,C\right)$ for a defined reference image $I_{\mathrm{ref}}$, which is typically the center image of the input bundle Ω. We assume that the input is calibrated, i.e., that the images are free of lens distortion, and that the full projection matrices $P_{k}=K\left[R\;t\right]$ are known.

Before any processing, a Gaussian image pyramid with *n* pyramid levels is computed for each image of the input bundle, allowing hierarchical processing. The lowest pyramid levels (l=0) contain the input images with their original image size. This results in an expansion of the input bundle Ω by n−1 additional sets. In the following, a superscript is used to mark the results and processes at a particular pyramid level. The pipeline is initialized at the level with the smallest image size and executes three successive computations at each pyramid level.

The first part of the actual processing, the depth estimation, computes a depth map $D^{l}$ and is in turn subdivided into a plane-sweep multi-image matching (Section 2.1), which generates depth hypotheses, and the SGM^x^ optimization (Section 2.2), which extracts the optimal depth from the set of hypotheses. The latter adopts the SGM algorithm [33,34] to the plane-sweep matching and extends it to account for non-fronto-parallel surface structures. A concluding depth refinement and median filter with a kernel size of 5×5 pixels is used to remove small outliers in the resulting depth map.

In the second and third computational parts of the hierarchical processing, a normal map $N^{l}$ (Section 2.3) and a confidence map $C^{l}$ (Section 2.4) are estimated from the previously computed depth map $D^{l}$. The confidence map contains pixel-wise confidence values in the interval $\left[0,1\right]$ with respect to the depth estimates. These confidence scores are computed based on the surface orientation at the considered pixel.

Inherent to a hierarchical coarse-to-fine processing, the depth map $D^{l}$ and the normal map $N^{l}$ computed at level *l* are used to initialize the depth map estimation at the next pyramid level l−1, as long as the lowest level of the image pyramid has not yet been reached, i.e., while l>0. Here, $D^{l}$ and $N^{l}$ are upscaled to the image size of the next pyramid level by nearest neighbor interpolation, yielding ${\bar{D}}^{l}$ and ${\bar{N}}^{l}$. Then, ${\bar{D}}^{l}$ is first used to compute the pixel-wise sampling range $\Gamma_{p}^{\;l-1}$ of the multi-image plane-sweep algorithm at the next pyramid level. Here, the $\Gamma_{p}^{\;l-1}$ is computed separately for each pixel p based on the previous depth estimate ${\bar{d}}_{\;p}^{\;l}={\bar{D}}^{\;l}\left(p\right)$ and a predefined window with a radius of Δd around ${\bar{d}}_{\;p}^{\;l}$:(1)$$\begin{array}{cc} \Gamma_{p}^{\;l-1} & =\left[d_{\;p,\min}^{\;l-1},d_{\;p,\max}^{\;l-1}\right],\mathrm{with}\\ d_{\;p,\min}^{\;l-1} & ={\bar{d}}_{\;p}^{\;l}-\Delta d,\\ d_{\;p,\max}^{\;l-1} & ={\bar{d}}_{\;p}^{\;l}+\Delta d. \end{array}$$

In the first iteration, the sampling range is set equally for all pixels and parameterized by the minimum and maximum scene depth: $\Gamma =\left[d_{\;\min},d_{\;\max}\right]$. The upscaled normal map ${\bar{N}}^{l}$ is used by one of the proposed SGM extensions to account for surface orientation within the scene. The final depth, normal, and confidence maps are the result of the processing at the lowest pyramid level. They are labeled D, N, and C, respectively, and have the same image size as the input images.

In a final post-processing step (Section 2.5), we use a Difference-of-Gaussian (DoG) filter [46], as well as a geometric filtering to remove remaining outliers by masking out regions with little image texture and enforcing geometric consistency.

### 2.1. Real-Time Dense Multi-Image Matching with Plane-Sweep Sampling

Given a multi-camera setup $c_{i}$ and an additional scene plane $\Pi =\left(n,\delta \right)$ positioned in the field of view of the cameras, the image point $p^{\mathrm{ref}}$ in the image of a preselected reference camera is mapped directly to the pixel $p^{k}$ in any other camera image via the homography H induced by the plane Π. The scene plane Π is parameterized by its normal vector n and its distance δ from the reference camera. Together with the corresponding camera poses, this homographic projection is formulated by:(2)$$\begin{array}{cc} p^{k} & =H\left(\Pi ,P_{\mathrm{ref}},P_{k}\right)\cdot p^{\mathrm{ref}}\;,\;\;\mathrm{with}\\ H\left(\Pi ,P_{\mathrm{ref}},P_{k}\right) & =K_{k}\cdot \frac{R-tn^{⊺}}{\delta }\cdot K_{\mathrm{ref}}^{-1}. \end{array}$$

Here, $K_{\mathrm{ref}}$ and $K_{k}$ denote the intrinsic matrices of both cameras, and $\left[R\;t\right]$ denotes the relative transformation matrix of the neighboring pose $P_{k}$ with respect to the reference pose $P_{\mathrm{ref}}$. As shown in Figure 2, Equation (Equation 2) is interpreted geometrically by casting a viewing ray through the pixel $p^{\mathrm{ref}}$ and intersecting it with the scene plane Π, yielding a scene point $x_{\Pi }$, which is then projected into the second camera, resulting in the image point $p^{k}$ [47].

#### 2.1.1. The Hierarchical Plane-Sweep Algorithm for Real-Time Multi-Image Matching

Based on the relationship between two or more cameras and a scene plane, Collins [26] proposed an algorithm for true multi-image matching. This algorithm samples the scene space between two bounding planes $\Pi_{\min}$ and $\Pi_{\max}$, located at $\delta_{\min}$ and $\delta_{\max}$, by sweeping a plane along its normal vector n through space and matching the input images according to Equation (Equation 2) for each distance $\delta \;\in \;\left[\delta_{\min},\;\delta_{\max}\right]$ of the plane relative to the reference camera. For each position of the plane, an arbitrary number of matching images are warped by the plane-induced homography $H_{\mathrm{ref}\to k}^{-1}$ into the view of the reference camera, where they are matched against the reference image. If the scene plane is close to a three-dimensional structure, then the corresponding image regions of the warped matching images overlap with those in the reference image, allowing the scene depth of the corresponding object to be derived from the parameterization of the corresponding plane (Figure 2). Initially referred to as the space-sweep algorithm, it has been adopted by numerous studies on multi-image matching and MVS [24,25,38], finally called the plane-sweep algorithm. This algorithm has proven to be very efficient in generating pixel-wise depth hypotheses and is therefore still widely used for the task of depth estimation [5,44,48] or novel view synthesis [49]. The presented hierarchical multi-image matching approach is based on the plane-sweep algorithm introduced by Pollefeys et al. [25] and described in Algorithm 1.


**Algorithm 1:** Plane-sweep multi-image matching executed at a specific pyramid level *l* of the proposed hierarchical processing scheme. 
 **Data:**a calibrated image bundle $\Omega^{l}$ at the pyramid level *l*, a set of planes Π with a normal vector n and varying distances δ as well as a local depth sampling range $\Gamma_{p}^{\;l}=\left[d_{\;p,\min}^{\;l},d_{\;p,\max}^{\;l}\right]$.

 **Result:**three-dimensional cost volume S, holding the pixel-wise matching score for each pixel $p^{\mathrm{ref}}\in I_{\mathrm{ref}}^{l}$ and plane Π.


**1**

determine bounding planes $\Pi_{\min}$ and $\Pi_{\max}$ located at $\delta_{\min}$ and $\delta_{\max}$, so that the local depth range $\Gamma_{p}^{\;l}$ is completely sampled (see Section 2.1.2).
**2**
**foreach** *pixel $p^{\mathrm{ref}}\in I_{\mathrm{ref}}^{l}$* **and** *distance $\delta \in \left[\delta_{\min},\delta_{\max}\right]$* **do** ((
**3**

Configure scene plane Π=(n,δ).
**4**

Determine pixels $p^{\;k}$ in all matching images $I_{k}^{l}\in \Omega^{l}\setminus I_{\mathrm{ref}}^{l}$: 

$$p^{\;k}=H\left(\Pi ,P_{\mathrm{ref}}^{l},P_{k}^{l}\right)\cdot p^{\mathrm{ref}}.$$


**5**

Warp local image patches $P_{k}^{l}\in I_{k}^{l}$ around $p^{\;k}$, with the same size as the support region of the matching cost function C(·), into $I_{\mathrm{ref}}^{l}$: 

$${\tilde{P}}_{k}^{l}=H{\left(\Pi ,P_{\mathrm{ref}}^{l},P_{k}^{l}\right)}^{-1}\cdot P_{k}^{l}.$$


**6**

Compute the matching cost $s\left(p,\Pi \right)$ between reference patch $P_{\mathrm{ref}}^{l}\in I_{\mathrm{ref}}^{l}$ and ${\tilde{P}}_{k}^{l}$ for left and right subset of cameras separately:


$$s^{L}\left(p,\Pi \right)=\sum_{k<\mathrm{ref}}C\left(P_{\mathrm{ref}}^{l},{\tilde{P}}_{k}^{l}\right),$$



$$s^{R}\left(p,\Pi \right)=\sum_{k>\mathrm{ref}}C\left(P_{\mathrm{ref}}^{l},{\tilde{P}}_{k}^{l}\right).$$


**7**

Store the minimum of left and right matching cost (accounting for occlusions as described by Kang et al. [50]) into three-dimensional cost volume S:


$$S^{l}\left(p,\Pi \right)=\min\left\{s^{L}\left(p,\Pi \right),s^{R}\left(p,\Pi \right)\right\}.$$


**8**

**end**

(




As part of the actual image matching, the Hamming distance of the census transform (CT) [51] and a negated, truncated and scaled form of the normalized cross-correlation (NCC) [38,41] are used and evaluated as cost functions C(·). And since the approach considers a bundle of input images with an equal number of matching images on either side of the reference image, the approach presented by Kang et al. [50] is adopted to account for occlusions, using the minimum aggregated matching cost of the left and right subset of matching images. The resulting three-dimensional cost volume $S^{\;l}$ is of size $w^{\;l}\times h^{\;l}\times \left|\delta^{\;l}\right|$, where $w^{\;l}$ and $h^{\;l}$ are the width and height of the reference image and $|\delta^{\;l}|$ is the number of plane positions at which the matching is performed, all with respect to the current pyramid level *l*. The cost volume $S^{\;l}$ is implemented as a dynamic cost volume [37] for all but the top pyramid level, since the sampling range $\Gamma_{p}^{\;l}$ is determined independently for each pixel p. Nevertheless, the complete set of plane distances $\delta \in [\delta_{\min},\delta_{\max}]$, deduced from Γ, are precomputed for each pyramid level *l* and are the same for all pixels. This in turn allows us to precompute the homographic mappings for all planes Π.

#### 2.1.2. Determining the Bounding Planes Corresponding to the Given Depth Range

As described before, it is assumed that two bounding planes, namely $\Pi_{\min}$ and $\Pi_{\max}$ with corresponding distances $\delta_{\min}$ and $\delta_{\max}$, between which the scene is to be sampled, are known. In the case of a fronto-parallel sampling strategy, i.e., $n={\left(0\;0\;-1\right)}^{⊺}$ with respect to the local camera coordinate system, the distances $\delta_{\min}$ and $\delta_{\max}$ are equal to the minimum and maximum depths, namely $d_{\min}$ and $d_{\max}$. This does not hold for non-fronto-parallel plane orientations. To find the bounding planes for slanted plane orientations, first a view-frustum is constructed, which corresponds to the reference camera for which the depth is to be estimated. This view-frustum is represented by a pyramid similar to the field of view of the camera, truncated by two fronto-parallel near and far planes located at $d_{\min}$ and $d_{\max}$. Given the four corner points of the view-frustum on the near plane $x_{i}^{\mathrm{near}}$ and the four on the far plane $x_{i}^{\mathrm{far}}$, the minimum and maximum distances $\delta_{\min}$ and $\delta_{\max}$ are determined as follows:(3)$$\begin{array}{cc} \delta_{\min} & =\min_{i}\left(\right|n^{⊺}\cdot x_{i}^{\mathrm{near}}\left|\right)\;,\;\;\mathrm{and}\\ \delta_{\max} & =\max_{i}\left(\right|n^{⊺}\cdot x_{i}^{\mathrm{far}}\left|\right). \end{array}$$

To avoid an orientation flip of the images, all camera centers $c_{i}$ must lie before $\Pi_{\min}$ with respect to the sweeping direction. Thus, for all cameras, $n^{⊺}\cdot c_{i}+\delta_{\min}>0$ must hold.

#### 2.1.3. Finding the Sampling Steps by Utilizing the Cross-Ratio

As stated by Equation (Equation 2), the sampling planes Π of the plane-sweep algorithm are parameterized by two parameters, namely the normal vector n, which denotes the orientation and the sweeping direction of the plane, and the orthogonal distance δ from the optical center of the reference camera $c_{\mathrm{ref}}$. The latter one determines the step size with which the scene is sampled. A simple approach would be to set the step size to sample the scene with a desired resolution, i.e., sweeping the planes at equidistant unit intervals through the scene space. However, there is no guarantee that a thorough sampling of the scene with a small step size will result in higher accuracy. If the step size is not chosen in accordance with the camera positions of the input images and the baseline between the cameras, the matching results of two or more successive plane positions may not reveal enough difference, thus introducing ambiguities between multiple plane hypotheses. Furthermore, for efficiency, it is important to vary the sampling rate in scene space with respect to the plane distance relative to the reference camera, since perspective projection requires an increasingly smaller step size as the plane moves closer to the camera.

Therefore, a common approach is to select the sampling positions of the planes according to the disparity change induced by two successive planes. The pixel-wise motion between the distorted images of two successive planes should not exceed an absolute value of 1 [25,52]. In this approach, we derive the distances of the sampling planes directly from the correspondences in image space by relying on the cross-ratio, which is invariant under perspective projection. Our approach to computing the distances δ of the sampling planes Π with respect to the reference camera was published in [11] and is illustrated in Figure 3 and summarized by Algorithm 2.
**Algorithm 2:** Finding plane distances δ by utilizing the cross-ratio.
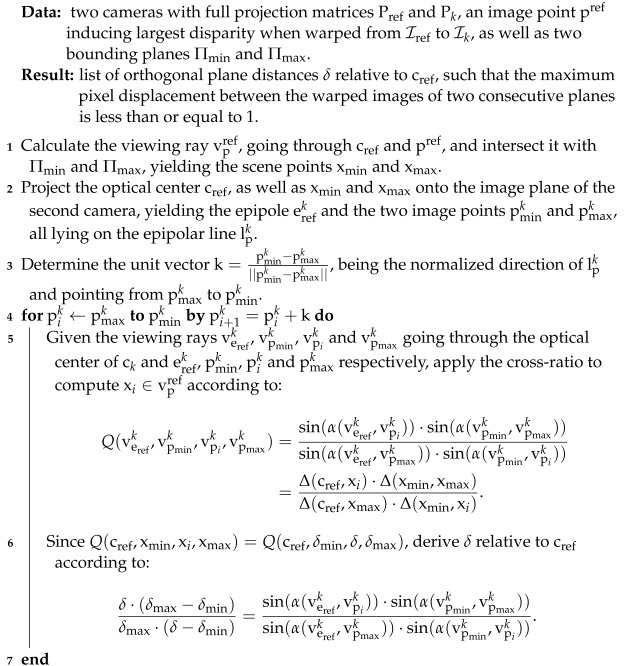


For $c_{k}$, we choose the camera that will induce the largest image offset, thus giving an upper bound on the disparity range. As noted by Pollefeys et al. [25], this is typically the camera farthest away from the reference camera. Similarly, we choose $p^{\mathrm{ref}}$ as the pixel that induces the largest disparity when warped from $I_{\mathrm{ref}}$ to $I_{k}$ via $H\left(\Pi_{\min},P_{\mathrm{ref}},P_{k}\right)$, typically one of the four corners. Furthermore, to account for all possible setups of $c_{\mathrm{ref}}$ and $c_{k}$, it is important to use $Q(v_{e_{\mathrm{ref}}}^{k},v_{p_{\min}}^{k},v_{p_{i}}^{k},v_{p_{\max}}^{k})$ in Algorithm 2, since $e_{\mathrm{ref}}^{k}$ would flip to the side of $p_{\max}^{k}$ if the focal plane of the reference camera is behind $c_{k}$. This approach is computationally efficient and is not restricted to a fronto-parallel orientation of the sampling planes, as long as the optical axis of the reference camera intersects the planes and the sweeping vector has a component that is parallel to the optical axis.

### 2.2. Depth Map Computation with Surface-Aware Semi-Global Matching

The hierarchical plane-sweep algorithm for multi-image matching produces a three-dimensional cost volume $S^{l}\left(p,\Pi \right)$ at each pyramid level, containing pixel-wise matching costs, given plane Π located at a distance δ orthogonal to the location $c_{\mathrm{ref}}$ of the reference camera. In the second stage of the depth estimation within FaSS-MVS, the cost volume is regularized by a semi-global optimization scheme, yielding a dense depth map $D^{l}$. It is based on the Semi-Global Matching (SGM) algorithm proposed by Hirschmüller [33,34] for the task of disparity estimation as part of the stereo normal case. It uses dynamic programming to efficiently minimize a two-dimensional Markov Random Field (MRF) energy function by aggregating the matching costs within the cost volume along numerous concentric one-dimensional paths.

Building on the original SGM approach, we propose three different optimization schemes (SGM^x^). Apart from a straightforward adaptation of the matching cost aggregation to plane-sweep sampling, we also adopt the approach of Scharstein et al. [41] to also favor slanted surfaces by taking into account surface information available in the form of surface normals. Furthermore, we investigate a third extension that penalizes deviations from the gradient of the minimum-cost path within the SGM optimization scheme. The subsequent extraction of the depth map D is performed analogously to the extraction of the disparity map within the SGM algorithm, where disparity is replaced by depth. If a fronto-parallel plane orientation is considered during the plane-sweep, the depth can be extracted directly from the plane parameterization. For non-fronto-parallel orientations, however, D is computed by a pixel-wise intersection of the viewing rays with the corresponding WTA solutions.

#### 2.2.1. Resolving Plane Hypotheses with Semi-Global Matching

Since the plane-sweep algorithm does not compute hypotheses on disparities, but rather pixel-wise plane distances relative to the reference camera and thus depth, the first SGM extension we propose is a straightforward adaption of the standard SGM algorithm to a multi-view plane-sweep sampling. In this, the formulation of the SGM path aggregation is modified to
(4)$$\begin{array}{c} L_{r}\left(p,\Pi \right)=S\left(p,\Pi \right)+\min_{\delta^{\prime }}\left(L_{r}\left(p-r,\Pi^{\prime }\right)+V_{\Pi }\left(\Pi ,\Pi^{\prime }\right)\right), \end{array}$$
where Π is the sampling plane at distance δ. The smoothness term $V_{\Pi }$ now penalizes the selection of different planes between adjacent pixels along the path $L_{r}$, instead of disparities. It is formulated as:(5)$$\begin{array}{cc} V_{\Pi }\left(\Pi ,\Pi^{\prime }\right) & =\left\{\begin{array}{cc} 0 & ,\;\mathrm{if}\;I\left(\Pi \right)=I\left(\Pi^{\prime }\right)\\ \varphi_{1} & ,\;\mathrm{if}\;\left|I\left(\Pi \right)-I\left(\Pi^{\prime }\right)\right|=1\\ \varphi_{2} & ,\;\mathrm{if}\;\left|I\left(\Pi \right)-I\left(\Pi^{\prime }\right)\right|>1, \end{array}\right. \end{array}$$
where I(·) is a function that returns the index of Π within the set of sampling planes (see Figure 4, Column 2). We denote this extension as *plane-wise* SGM (SGM^Π^). In our previous publication [12], we have referred to this extension as *fronto-parallel* SGM (SGM^fp^), since we have only considered a fronto-parallel sweeping direction so far. However, the extension is not restricted to a fronto-parallel plane orientation in the plane-sweep sampling and will also be evaluated with slanted planes in the scope of this work. Given a pixel-wise WTA plane parameterization, the corresponding depth is extracted by intersecting the viewing ray through pixel $p={\left(p_{x}\;p_{y}\right)}^{⊺}$ with the corresponding plane:(6)$$\begin{array}{c} d_{p}=\frac{-\delta }{n^{⊺}\cdot K^{-1}\cdot {\left(p_{x}\;p_{y}\;1\right)}^{⊺}}. \end{array}$$

#### 2.2.2. Incorporating Surface Normals to Adjust the Zero-Cost Transition

The smoothness term of the initial SGM algorithm is formulated with discrete disparity differences, penalizing discrete disparity jumps between neighboring pixels. In its optimization scheme, it does not consider subpixel disparities and thus favors fronto-parallel surface structures, leading to staircase artifacts if no post-processing is applied [41]. The same applies to our first extension, SGM^Π^. Although plane-sweep sampling also supports non-fronto-parallel plane orientations, the smoothness term of SGM^Π^ (see Equation (Equation 5)) does not, and strongly penalizes index jumps in the sampling planes greater than 1. While this is desired if the plane orientation coincides with the surface orientation, it will still lead to staircasing artifacts if the surface and plane orientations do not align. To overcome the favoring of fronto-parallel structures and to adjust the smoothness term of SGM to surfaces that are slanted with respect to the sampling direction, Scharstein et al. [41] suggest adding an offset to the smoothness term. This offset can be extracted from additional information about the surface orientation, e.g., surface normals, which will make the zero-cost transition coincide with the surface orientation. We adopt this approach as part of our second extension, and thus call it *surface normal* SGM (SGM^Π-sn^).

In our hierarchical approach, we extract the normal vectors from the normal map $N^{l+1}$, which was estimated in the previous level of the pyramid (see Figure 1). The pixel-wise normal vectors $n_{x}=N^{l+1}\left(p\right)$ indicate the surface orientation at the scene point x, which is computed by intersecting the viewing ray through p with the plane Π. From this, the discrete index jump ${\Delta i}_{\mathrm{sn}}$ through the set of sampling planes can be calculated, which is caused by the tangent plane to $n_{x}$. Since the plane-sweep sampling is not restricted to fronto-parallel plane orientations, the index jump ${\Delta i}_{\mathrm{sn}}$ must be calculated based on the difference between the tangent plane at $x_{\Pi }$ and the orientation of the sampling planes in the direction r of the currently considered aggregation path. With ${\Delta i}_{\mathrm{sn}}$, the smoothness term used by our extension SGM^Π-sn^ is adjusted according to
(7)$$\begin{array}{cc} V_{\Pi \text{-}sn}\left(\Pi ,\Pi^{\prime }\right) & =\left\{\begin{array}{cc} 0 & ,\;\mathrm{if}\;I\left(\Pi \right)+{\Delta i}_{\mathrm{sn}}=I\left(\Pi^{\prime }\right)\\ \varphi_{1} & ,\;\mathrm{if}\;\left|I\left(\Pi \right)+{\Delta i}_{\mathrm{sn}}-I\left(\Pi^{\prime }\right)\right|=1\\ \varphi_{2} & ,\;\mathrm{if}\;\left|I\left(\Pi \right)+{\Delta i}_{\mathrm{sn}}-I\left(\Pi^{\prime }\right)\right|>1, \end{array}\right. \end{array}$$

This allows the zero-cost transition of the SGM path aggregation to be aligned with the surface orientation of the scene (see Figure 4, Column 3). The pixel-wise discrete index jumps can be computed once for each pixel p and each path direction r, as also noted by Scharstein et al. [41], with little computational overhead.

#### 2.2.3. Penalizing Deviations from the Gradient of the Minimum-Cost Path

Instead of relying on additional information, e.g., normal vectors, the third of our proposed extensions computes the running gradient ∇r of the minimum-cost path in scene space in order to adjust the zero-cost transition in the aggregation of path costs. Hence, it is denoted as *path gradient* SGM (SGM^Π-pg^).

The gradient vector $\nabla r=x-x^{\prime }$ in scene space is computed dynamically while traversing the path r. Again, x is the scene point found by intersecting the viewing ray through p with Π, while $x^{\prime }$ is the scene point parameterized by $p^{\prime }$ and the plane ${\hat{\Pi }}^{\prime }$. Here, $p^{\prime }=p+r$ represents the predecessor of p along the path r and ${\hat{\Pi }}^{\prime }$ denotes the plane at distance $\hat{\delta }={arg min}_{\delta }L_{r}\left(p^{\prime },\Pi \right)$ associated with the previous minimum costs.

From this, a discrete index jump ${\Delta i}_{\mathrm{pg}}$ is computed, which is again used to account for possibly slanted surfaces in scene space by adjusting the zero-cost transition of the smoothness term according to
(8)$$\begin{array}{cc} V_{\Pi \text{-}pg}\left(\Pi ,\Pi^{\prime }\right) & =\left\{\begin{array}{cc} 0 & ,\;\mathrm{if}\;I\left(\Pi \right)+{\Delta i}_{\mathrm{pg}}=I\left(\Pi^{\prime }\right)\\ \varphi_{1} & ,\;\mathrm{if}\;\left|I\left(\Pi \right)+{\Delta i}_{\mathrm{pg}}-I\left(\Pi^{\prime }\right)\right|=1\\ \varphi_{2} & ,\;\mathrm{if}\;\left|I\left(\Pi \right)+{\Delta i}_{\mathrm{pg}}-I\left(\Pi^{\prime }\right)\right|>1, \end{array}\right. \end{array}$$

This implicitly penalizes deviations from the running gradient between two scene points corresponding to two consecutive pixels on the aggregation path r (see Figure 4, Column 4).

### 2.3. Extraction of Surface Normals from Depth Maps

From the estimated depth map D, our approach computes a normal map N, which holds the local surface orientations in the form of three-dimensional normal vectors. The surface normal vectors $n_{p}=h_{p}\times v_{p}$ are computed using the cross-product, where $h_{p}$ is the difference vector between the reprojected scene points of two neighboring pixels to p in horizontal direction and $v_{p}$ is the difference vector in vertical direction.

Using only the cross-product to compute the surface orientation does not include any local smoothness assumption. Therefore, we use an appearance-based weighted Gaussian smoothing in a local two-dimensional window $W_{p}$ around p, which adjusts the smoothing strength depending on the intensity difference between $q\in W_{p}$ and p:(9)$$\begin{array}{c} N\left(p\right)=\frac{{\bar{n}}_{p}}{\left|{\bar{n}}_{p}\right|}, \end{array}$$
with
(10)$${\bar{n}}_{p}=n_{p}+\sum_{q\in W_{p}}n_{q}\cdot \frac{1}{\sqrt{2\pi \sigma^{2}}}\cdot \exp\left(-\frac{{\left(q-p\right)}^{2}}{2\sigma^{2}}-\frac{\Delta I_{\mathrm{pq}}}{\beta }\right).$$

In this, β is set to 10, while σ is fixed to the radius of $W_{p}$.

### 2.4. Estimation of Confidence Measures Based on Surface Orientation

Besides the depth map D and the normal map N, the presented approach also computes confidence measures for the depth estimates in the range of $\left[0,\;1\right]$ and stores them in a confidence map C. Such confidence measures allow subsequent reasoning about the certainty of the corresponding estimates and thus improve further processing. Thus, confidence maps are useful by-products for subsequent steps such as depth map fusion or scene interpretation. Furthermore, they allow us to gain more insight into the effects of different configurations of the presented approach.

The computation of the pixel-wise confidence measures is based on the geometric properties of the estimated depth map and is derived from the normal vectors stored inside the normal map N and the plane orientations of the plane-sweep sampling. In particular, the geometric confidence measure is based on the enclosed angles between the local surface orientation stored inside the normal map $n_{p}=N\left(p\right)$, the orientation of the sampling plane $n_{\Pi }$, and the inverted viewing direction v. This is taken from the geometric weighting factor proposed by Kolev et al. [53]. They argue that a depth estimate is more accurate when the surface orientation of the observed geometry is fronto-parallel to the image plane of the camera, and less accurate when the camera observes slanted surfaces. This correlation is modeled by the scalar product between the surface orientation and the inverted viewing direction. Since image warping, as part of image matching, can be aligned with the surface orientation by adjusting the normal vector of the plane-sweep algorithm, the plane orientation is also taken into account. Thus, the geometry-based weighting factor is calculated as follows:(11)$$\begin{array}{cc} C(p) & =\left\{\begin{array}{cc} \frac{\langle n_{p},n_{\Pi }\rangle \langle n_{\Pi },v\rangle \;-\;\cos\rho }{1\;-\;\cos\rho } & ,\;\mathrm{if}\;\{∢\left(n_{p},n_{\Pi }\right)\wedge ∢\left(n_{\Pi },v\right)\}\leq \rho \\ 0 & ,\;\mathrm{otherwise}\;. \end{array}\right. \end{array}$$

All of the above vectors are assumed to be normalized and given with respect to the local coordinate system of the camera; thus, $v={\left(0\;0\;-1\right)}^{⊺}$. As in the work of Kolev et al. [53], a critical angle $\rho =60^{\circ }$ is used to mark the measurements, for which the enclosed angles exceed this threshold, as unreliable. The additional consideration of $n_{\Pi }$ in Equation (Equation 11) implicitly models the indirect matching of the input images via the plane-induced homography.

### 2.5. Post-Processing and Depth Map Filtering

In a final post-processing step, remaining outliers are removed from the depth, normal and confidence maps by applying a Difference-of-Gaussian (DoG) filtering (Section 2.5.1) and an outlier removal based on geometric consistency (Section 2.5.2).

#### 2.5.1. Difference-of-Gaussian Filtering

As proposed by Wenzel [46], the DoG filter allows us to remove estimates of D, N, and C by masking pixels in image regions that provide little textural information (e.g., blurred or overexposed areas). It is assumed that image matching in such regions is ambiguous and leads to less accurate results. The DoG filter is used to detect weakly textured areas within the reference image $I_{\mathrm{ref}}$ and to build a binary image mask that is used to remove the estimates from the corresponding maps. Algorithm 3 provides an overview of the implementation of the DoG filter used, which is similar to the one proposed in [46].
**Algorithm 3:** The Difference-of-Gaussian filter to invalidate all image pixels belonging to weakly textured areas.**Data:** unfiltered depth, normal and confidence maps (D, N and C) as well as corresponding reference image $I_{\mathrm{ref}}$.**Result:** filtered D, N and C, in which all estimates corresponding to weakly-textured areas in $I_{\mathrm{ref}}$ are removed.**1**Use a Gaussian filter with a kernel of 7×7 pixels to smooth the reference frame $I_{\mathrm{ref}}$, yielding $I_{\mathrm{ref}}^{\mathrm{smooth}}$.**2**Compute the DoG image depicting local image gradients, according to: $I_{\mathrm{ref}}^{\mathrm{DoG}}=I_{\mathrm{ref}}-I_{\mathrm{ref}}^{\mathrm{smooth}}$.**3**Apply a binary threshold to compute the DoG mask $M^{\mathrm{DoG}}$, marking all image areas in which the intensity change is greater than 0.5.**4**Remove activation areas smaller than 7 pixels in $M^{\mathrm{DoG}}$ by applying a speckle filter.**5**Dilate $M^{\mathrm{DoG}}$ with a kernel size of 3×3 pixels to fill small holes in activation areas.**6**Remove deactivation areas smaller than 21 pixels by applying a speckle filter to the inverted DoG mask $M^{DoG\text{-}inv}=1-M^{\mathrm{DoG}}$.**7**Invalidate pixels in D, N and C for which $M^{\mathrm{DoG}}=1$.

#### 2.5.2. Geometric Consistency Based on Mutual Reprojection Error

If multiple depth maps $D_{k}$ with corresponding projection matrices $P_{k}$ are available, e.g., when performing reconstruction by MVS or when considering a sequence of images as input and a temporal consistency is to be established, a geometric consistency check can be performed by relying on the mutual reprojection error. As formulated by Schönberger et al. [4], each pixel $p^{\mathrm{ref}}$ of a selected reference depth map $D_{\mathrm{ref}}$ with a depth estimate $d_{p}^{\;\mathrm{ref}}$ is projected into the view of another depth map $D_{k}$ by $H_{p}$, according to $d_{p}^{\;\mathrm{ref}}$ and the corresponding projection matrices $P_{\mathrm{ref}}$ and $P_{k}$, resulting in the image point $p^{k}$. Given $p^{k}$ and the corresponding depth $d_{p}^{\;k}$ from $D_{k}$, the image point $p^{k}$ is projected back into the view of $D_{\mathrm{ref}}$ by $H_{p}^{k}$, resulting in ${\tilde{p}}^{\mathrm{ref}}$. Finally, if the Euclidean distance between $p^{\mathrm{ref}}$ and ${\tilde{p}}^{\mathrm{ref}}$, i.e., the reprojection error $\epsilon_{r}^{k}\left(p\right)$, exceeds a given threshold $\eta_{r}$, the estimate at $p^{\mathrm{ref}}$ is invalidated.

We adopt this approach to perform a final geometry-based filtering between a set of depth maps within a sliding window. This is not part of the actual hierarchical processing pipeline, but rather a separate post-processing step, since it requires the results of other image bundles of the input sequence. If possible, the middle depth map of the sliding window Ψ is chosen as the reference view on which the filtering is performed. At the beginning or end of the sequence, where the sliding window would exceed the boundaries, the window is shifted to either side of the reference view, so that it is always within the boundaries of the sequence and no depth map is filtered multiple times. Besides the threshold of the reprojection error $\epsilon_{r}^{k}$, another criterion is introduced to evaluate the geometric consistency, namely the number of neighboring views for which the reprojection error is within the threshold, i.e., the number of hits: $\epsilon_{h}\left(p\right)=\sum_{k}\left[\epsilon_{r}^{k}\left(p\right)<\eta_{r}\right]$, where [·] is the Iverson bracket. Algorithm 4 gives an overview of geometric consistency and the corresponding filtering of the depth, normal, and confidence maps. In this work, the sliding window size is empirically set to |Ψ|=5, the reprojection error threshold to $\eta_{r}=10$, and the consistency threshold to $\eta_{h}=3$.
**Algorithm 4:** The geometric consistency filter for multiple depth maps.
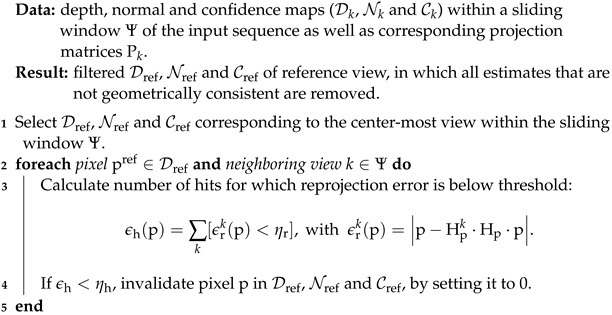


### 2.6. Evaluation Datasets

The presented approach is quantitatively evaluated on two public datasets, namely the DTU Robot MVS dataset [54,55] and the 3DOMcity Benchmark dataset [56], which also provide appropriate ground truth. These two datasets are based on images of scale modeled buildings and an urban landscape from which an accurate ground truth is acquired. For a qualitative evaluation and discussion of the applicability of the presented approach for online dense image matching and 3D reconstruction, two privately captured datasets of real-world scenes are used, hereafter referred to as the TMB and FB datasets. In the following, the characteristics of these datasets are briefly introduced. In particular, we discuss which parts of the datasets are used and what kind of ground truth is available for the evaluation. The key characteristics are summarized in Table 1.

#### 2.6.1. DTU Robot MVS Dataset

The DTU Robot MVS dataset (Figure 5, Column 1) consists of 124 different tabletop scenes, of which we used 21 scans of different building models, as these scenes are closest to the target use case. For each scene, there are input images taken from 49 locations distributed in an orbital pattern around the tabletop scene. In addition, a ground truth is provided for each scene in the form of a detailed point cloud captured by a structured-light scanner. For the quantitative evaluation, the already undistorted images with a resolution of 1600×1200 pixels, together with the provided intrinsic and extrinsic camera projection matrices, were used as input data for the approach. Since the focus of this work is on the estimation of depth and normal maps only, corresponding ground truth data are rendered from the detailed point cloud data.

#### 2.6.2. 3DOMcity Benchmark Dataset

Depending on the aircraft and its environment, an orbital trajectory as shown by the DTU benchmark data may not always be feasible or desirable. The data provided as part of the 3DOMcity Benchmark [56] (Figure 5, Column 2), however, simulate a grid flight where the aircraft flies linearly over the area of interest with a fixed camera orientation relative to the sensor carrier. In this case, images of a scaled urban scene consisting of buildings of various sizes and shapes, as well as roads and vegetation, are captured with a DSLR camera that is moved in parallel lines over the model along a rigid bar. To use the data of the 3DOMcity Benchmark for a quantitative evaluation of the performance of the presented approach, the already undistorted images are first downscaled to a size of 1798×1200 pixels, preserving the initial aspect ratio, before the intrinsic camera parameters are estimated with the help of COLMAP [3]. The extrinsic camera data are extracted from the reference provided as part of the benchmark. To evaluate the accuracy of the depth maps, the reference point cloud computed by the semi-global dense image matching (DIM) algorithm is rendered from the viewpoints of the input images, as in the case of the DTU Benchmark dataset.

#### 2.6.3. Real-World Use-Case-Specific TMB and FB Datasets

The strength and purpose of the DTU and 3DOMcity benchmark datasets is their small size and the associated ability to record or compute accurate reference data, which in turn facilitates a quantitative evaluation of the accuracy of the evaluated algorithms. However, these datasets were recorded in controlled environments and do not fully address the use case targeted by the presented approach. In order to perform a qualitative evaluation on real data, appropriate test data were collected in private datasets.

This dataset is two-fold. The first part, the TMB dataset (Figure 5, Column 3), consists of four sequences captured by a DJI Phantom (DJI, Shenzhen, China) 3 Professional flying around a freestanding house and containers at altitudes between 8 m and 15 m. The second part, referred to as the Fire Brigade (FB) dataset (Figure 5, Column 4), was captured during a fire training exercise around a large industrial building. The data were collected using a DJI Matrice 200 with a Zenmuse XT2 sensor flying linearly over the area where the exercise was conducted. For all sequences, images were captured at a frame rate of approximately 1 FPS and downsampled to an image size of 1920×1080 pixels. Images that are not suitable as input for the presented approach, e.g., by providing too little offset, are discarded.

### 2.7. Error Measures

To directly quantify error between the estimated depth map $D_{\mathrm{est}}$ and the corresponding ground truth $D_{\mathrm{gt}}$ during the experiments, absolute and relative L1 measures are used:(12)$$\begin{array}{c} L1\text{-}abs\left(D_{\mathrm{est}},D_{\mathrm{gt}}\right)=\frac{1}{\left|V\right|}\sum_{p\in V}|D_{\mathrm{est}}\left(p\right)-D_{\mathrm{gt}}\left(p\right)|\;,\;\;\mathrm{and}\; \end{array}$$
(13)$$\begin{array}{c} L1\text{-}rel\left(D_{\mathrm{est}},D_{\mathrm{gt}}\right)=\frac{1}{\left|V\right|}\sum_{p\in V}\frac{|D_{\mathrm{est}}\left(p\right)-D_{\mathrm{gt}}\left(p\right)|}{D_{\mathrm{gt}}\left(p\right)}\;.\;\;\;\;\;\;\;\; \end{array}$$

Here, V denotes the set of pixels for which both $D_{\mathrm{est}}$ and $D_{\mathrm{gt}}$ have valid depth measurements. While L1−abs provides an absolute and thus interpretable insight into the mean error of the estimated depth map, it is rather unsuitable for comparing results across multiple datasets with different depth ranges. This is because the error of depth measurements typically increases with depth, resulting in a higher absolute error for datasets with greater scene depth. To compensate for this effect, the relative L1−rel measure normalizes the absolute difference by the depth stored at the corresponding ground truth pixel. This reduces the effect that erroneous pixels in distant areas of the scene have on the error score, while increasing the weight of pixels close to the camera.

The two error measures introduced above provide a simple strategy for evaluating the error of the estimates. However, they do not allow one to reason about the completeness and density of the estimated depth map. Since the focus of this work is on dense MVS, it is also of great interest to know how many pixels of $D_{\mathrm{est}}$ are actually filled with correct estimates. Two closely related error measures are used for this, namely the accuracy (${\mathrm{Acc}}_{\theta }$) and the completeness (${\mathrm{Cpl}}_{\theta }$). These scores are typically used to evaluate classification tasks, but in recent years, they have also been used to evaluate range measurements [57,58]. On the one hand, the accuracy ${\mathrm{Acc}}_{\theta }$ indicates the number of pixels within the estimated depth map $D_{\mathrm{est}}$ for which the corresponding depth value is within a given threshold θ to the ground truth:(14)$${\mathrm{Acc}}_{\theta }\left(D_{\mathrm{est}},D_{\mathrm{gt}}\right)=\frac{1}{\left|E\right|}\sum_{p\in V}\left[\max\left(\frac{D_{\mathrm{est}}\left(p\right)}{D_{\mathrm{gt}}\left(p\right)},\frac{D_{\mathrm{gt}}\left(p\right)}{D_{\mathrm{est}}\left(p\right)}\right)<\theta \right]\;.$$

The completeness ${\mathrm{Cpl}}_{\theta }$, on the other hand, indicates the fraction of the ground truth pixels for which estimates exist that are within the given distance threshold to the reference:(15)$${\mathrm{Cpl}}_{\theta }\left(D_{\mathrm{est}},D_{\mathrm{gt}}\right)=\frac{1}{\left|G\right|}\sum_{p\in V}\left[\max\left(\frac{D_{\mathrm{est}}\left(p\right)}{D_{\mathrm{gt}}\left(p\right)},\frac{D_{\mathrm{gt}}\left(p\right)}{D_{\mathrm{est}}\left(p\right)}\right)<\theta \right]\;.$$

Again, V holds the set of pixels for which both $D_{\mathrm{est}}$ and $D_{\mathrm{gt}}$ have valid depth measurements. Similarly, E denotes the set of pixels with valid estimates, while G holds the pixels with valid ground truth values. In both Equations (Equation 14) and (Equation 15), the operator [·] refers to the Iverson bracket. The threshold θ is given as a percentage of the corresponding ground truth value. For example, ${\mathrm{Acc}}_{1.25}$ and ${\mathrm{Cpl}}_{1.25}$ give the fraction of pixels with respect to the $D_{\mathrm{est}}$ and $D_{\mathrm{gt}}$ for which the difference between the estimate and the ground truth is less than 25% of the corresponding ground truth depth. These two measures are combined into a single score, the $F_{\theta }$ score, which is the harmonic mean of ${\mathrm{Acc}}_{\theta }$ and ${\mathrm{Cpl}}_{\theta }$:(16)$$F_{\theta }\left(D_{\mathrm{est}},D_{\mathrm{gt}}\right)=2\cdot \frac{{\mathrm{Acc}}_{\theta }\cdot {\mathrm{Cpl}}_{\theta }}{{\mathrm{Acc}}_{\theta }+{\mathrm{Cpl}}_{\theta }}\;.$$

Thus, a high $F_{\theta }$-score indicates a good trade-off between the achieved accuracy of the depth map and its completeness with respect to the ground truth.

## 3. Results

The following sections present the results of the experiments conducted. They evaluate and analyze different aspects of the presented approach, such as accuracy, efficiency, and application-specific usability. First, the chosen configuration of the hyperparameters, i.e., those of the hierarchical processing scheme, similarity metrics and cost function, is outlined in Section 3.1. In Section 3.2, the ability of the three SGM extensions to reconstruct non-fronto-parallel surface structures is evaluated and compared to the effects of using a non-fronto-parallel plane orientation within plane-sweep sampling. An evaluation of the improvements obtained by post-filtering is presented in Section 3.3. In Section 3.4 and Section 3.5, FaSS-MVS is evaluated and compared with related approaches from the literature, both in terms of accuracy and runtime. Finally, the results of use-case-specific experiments are presented and qualitatively illustrated in Section 3.6.

The entire processing pipeline of the presented approach, except for the generation of the Gaussian image pyramids and the parameterization of the plane-sweep algorithm, is implemented in CUDA and thus optimized for massively parallel computing by general purpose computation on a GPU (GPGPU), which in turn is embedded in a C++ application. All experiments and timing measurements were performed on an NVIDIA Titan X (Santa Clara, CA, USA) GPU and an Intel XEON CPU E5-2650 (Santa Clara, CA, USA) running at 2.20 GHz. Although the CPU is designed for a server architecture, only a small part of our approach is run on the CPU, and thus its superiority over commodity desktop hardware is insignificant.

### 3.1. Configuration of the Hierarchical Plane-Sweep Dense Multi-Image Matching

To find the best parameterization for the hierarchical DIM, i.e., the optimal number of pyramid levels, the best similarity measure and cost function for the plane-sweep DIM, as well as the appropriate plane orientation, we performed several ablation studies as described in Appendix A, Appendix B, and Appendix C, respectively. In summary, the input bundle size is set to |Ω|=5 and the pyramid height is set to n=3 for the DTU dataset and n=2 for the 3DOMcity dataset. A fronto-parallel plane orientation, i.e., $n={\left(0\;0\;-1\right)}^{⊺}$, is used for the plane-sweep sampling. As a similarity measure and cost function in the DIM, the truncated, inverted and scaled NCC with a support region of 5×5 pixels is used. Although the NCC with a support region of 9×9 pixels achieves the best results, ${\mathrm{NCC}}_{5\times 5}$ is chosen for further experiments, since the error increase is small, but the computational complexity is significantly lower and the throughput higher than that of ${\mathrm{NCC}}_{9\times 9}$ as measured by Ruf et al. [36]. Based on the chosen cost function, we set the SGM penalty $\varphi_{1}$ to 100. To preserve depth discontinuities at object boundaries, we adaptively adjust the second penalty $\varphi_{2}$ based on the absolute intensity difference between two neighboring pixels, as formulated by Scharstein et al. [41], with α=8 and β=10. And since the presented approach uses multiple matching images, the SGM penalties are multiplied by the number of input images within the left and right subsets with respect to $I_{\mathrm{ref}}$, since the matching costs are summed within these image sets. These hyperparameters will be used for all subsequent experiments.

### 3.2. Evaluation of the Surface-Aware Extensions to SGM

As described in Section 2.2, in addition to the straightforward combination of SGM with plane-sweep sampling (SGM^Π^), this work includes two further surface-aware extensions to SGM, namely the incorporation of surface normals to adjust the zero-cost transition in SGM path aggregation (SGM^Π-sn^) and the penalization of deviations from the gradient of the minimum-cost path (SGM^Π-pg^). In the following, the results obtained by SGM^Π-sn^ and SGM^Π-pg^ in combination with a fronto-parallel sampling plane orientation are evaluated and compared with those obtained by SGM^Π^.

The quantitative results presented in Table 2 show only small differences in the L1 error between the different implementations of the SGM optimization. While for the DTU dataset, the best results are achieved by the SGM^Π-pg^ implementation, for the 3DOMcity dataset, the standard adaptation of the SGM optimization to plane-sweep sampling, i.e., SGM^Π^, achieves the lowest error. The relative L1 error shows no difference. This is due to the fact that the individual L1−rel values only start to differ after the fourth decimal place. Nevertheless, the ranking of the L1−rel scores is the same as that of the L1−abs scores. In a qualitative comparison, Figure 6 shows that SGM^Π-sn^ leads to a seemingly smoother depth and normal map (e.g., on the ground plane), but at the same time loses small details and increases unwanted depth discontinuities in some areas, such as the building facade. A close comparison of the normal maps between SGM^Π^ and SGM^Π-pg^ shows slightly less staircasing artifacts in the case of SGM^Π-pg^, which also supports the slightly lower error in Table 2. However, a qualitative comparison of the results on the 3DOMcity dataset in Figure 7 does not show any noticeable differences between the different implementations. The reason for the small L1−abs error achieved by SGM^Π^ on the 3DOMcity dataset is thought to be due to the fact that the 3DOMcity dataset also contains a subset of nadir images in which there are few slanted surfaces and the fronto-parallel orientation of the sampling planes coincides with most of the scene structure.

To further quantify the strengths and weaknesses of the three different SGM aggregation strategies, three receiver operating characteristic (ROC) curves, one for each extension, are plotted for each dataset in Figure 8. These curves illustrate the error rate achieved by the corresponding SGM extension as a function of increasing density of the estimated depth map. The density of the depth map is varied by sampling the number of pixels in steps of 5% based on their ordered confidence stored in C, going from a high to a low confidence estimate. The average error rate is quantified by $1\text{-}{Acc}_{1.05}$ (see Equation (Equation 14)) and indicates the number of sampled pixels in D whose absolute difference from the ground truth exceeds 5% of the ground truth value. Thus, at a low density of D, i.e., a high confidence threshold, the error rate should ideally be at its minimum and then increase with increasing density, reaching the total error of D at a density of 100%. The plots start at a density of 5%, since the error rate at a density of 0% is undefined. However, analyzing the ROC curves of each method individually is not very meaningful. So in Table 3, we also provide data on the area under curve (AUC) along with the optimal AUC (AUC-Opt.) and the difference between the two (ΔAUC) for the three different SGM implementations, as discussed by Mehltretter and Heipke [59], to quantitatively assess the accuracy of the estimated depth and confidence map. Since the AUC-Opt. represents the area under curve for an optimal confidence map, the smaller the difference ΔAUC, the more accurate the confidence map. The curves in Figure 8 as well as the results in Table 3 support the superiority of the surface-aware SGM extensions over the standard SGM adaptation to plane-sweep sampling. For both datasets, the curves and the difference of the AUC to the AUC-Opt. of SGM^Π-sn^ and SGM^Π-pg^ are lower than those of SGM^Π^, indicating lower error rates. However, the fact that most of the ROC curves start with a high error rate at a density of 5%, and then drop before rising again, suggests that the estimated confidence values do not adequately represent the certainty of the depth estimates. The reasons for this are many and will be discussed further in Section 4.4.

### 3.3. Improvements Gained by Post-Filtering

In the following, the effects of the implemented post-filtering methods to remove remaining outliers and supposedly wrong estimates by Difference-of-Gaussian (DoG) filtering (see Section 2.5.1) and a geometric consistency check (GCC) (see Section 2.5.2) are examined. While the latter relies on the actual estimates, the DoG filter is based on the assumption that image regions with low texture could lead to ambiguities in the image matching and, in turn, incorrect estimates. However, this can lead to the erroneous removal of good or even correct estimates.

Instead of using the absolute and relative L1 metrics to quantitatively evaluate the results achieved when using post-filtering, the effects are evaluated using the accuracy measure ${\mathrm{Acc}}_{\theta }$ (see Equation (Equation 14)) and the completeness measure ${\mathrm{Cpl}}_{\theta }$ (see Equation (Equation 15)). This is because they indirectly contain information about the density of the resulting depth maps, which should ideally be as high as possible. Since the individual sequences of the 3DOMcity dataset consist of too few images to perform a GCC with the parameterization mentioned in Section 2.5.2, this experiment is only performed on the DTU benchmark dataset. Figure 9 shows the results of different post-filtering strategies, i.e., DoG filtering, GCC, and a combination of both, performed in combination with the three different SGM extensions and a fronto-parallel sampling. For reference, the accuracy–completeness curves resulting from the corresponding configurations without post-filtering are also shown. When constructing the curves, the threshold θ is varied within the list of {1.25,1.20,1.15,1.10,1.05,1.01}. Note that, as the threshold decreases, the accuracy and completeness rates also decrease. The highest values are obtained with θ=1.25.

Most evidently, Figure 9 again shows that there is not much difference in the overall error between the three SGM implementations. However, the accuracy–completeness curves clearly show the differences between the post-filtering strategies. Unsurprisingly, the reference configuration with no filtering achieves the highest completeness, since no estimates are removed from the predicted depth map, which results in the lowest accuracy. The use of DoG filtering significantly improves this, as it is likely to remove a significant number of false estimates from poorly textured areas. However, as expected, the DoG filter probably also removes a number of correct estimates, as the use of filtering based on geometric consistency achieves a similar completeness, but with a higher accuracy. In particular, looking at the values for θ=1.01, i.e., the lower left end of each curve, the use of a GCC achieves an increase in completeness of about 5%, while exceeding the accuracy of the DoG filter by more than 10%. However, a clear recommendation as to which filter to use cannot be made, since both filtering strategies have their strengths and weaknesses, especially with respect to online processing, as discussed in Section 4.3. A combination of both filters is not motivated. Although the accuracy increases slightly, the completeness decreases by more than 20% in some cases. Moreover, this effect can also be achieved by lowering the threshold of the reprojection error $\eta_{r}$ in the geometric consistency check, which will probably increase the accuracy even more.

Finally, to directly compare the different SGM extensions in combination with the GCC that gives the best results, the corresponding F-scores (see Equation (Equation 16)) for each evaluated θ are listed in Table 4. Just like the results shown in Table 2, the F-scores reveal the superiority of SGM^Π-pg^ over the other two implementations, since for all θ but one, SGM^Π-pg^ achieves the highest F-score.

### 3.4. Comparison to Related Approaches from Literature

Based on the previous experiments and the knowledge gained about the best performing configuration, we now perform a series of experiments on the DTU dataset to compare FaSS-MVS with related approaches from the literature. On the one hand, we compare our results with those of a related approach for online dense MVS, namely the PlaneSweepLib (PSL) [31], which is also used by OpenREALM [6]. The algorithm provided by the PSL is very similar to ours, but does not have hierarchical processing and does not perform post-processing based on geometric queues. In addition, the PSL uses a Bayesian formulation to extract the depth map from the generated depth hypotheses, while FaSS-MVS relies on optimizing an MRF using dynamic programming. In the following experiments, we configure the PSL to also use five input images, 128 planes to generate depth hypotheses, the NCC as similarity measure, and the reference split [50] to account for occlusions. We compare the performance of the PSL with different sized support regions for the NCC, namely with a neighborhood size of 5×5 pixels and 11×11 pixels. And since the PSL does not include a geometric verification of the estimates, we also combine it with the filtering based on GCC in the same configuration as described above.

We also evaluate FaSS-MVS against two offline MVS approaches, namely the widely used and open source COLMAP toolbox and the more recent ACMMP [18]. While COLMAP provides the complete reconstruction pipeline, i.e., including the estimation of camera poses by SfM and the fusion of the depth maps into a 3D point cloud, only the geometric depth maps estimated by the provided MVS approach [4] with the default configuration are used for comparison. Like many other MVS techniques, COLMAP as well as FaSS-MVS and PSL have difficulty estimating reliable pixel correspondences and thus depth values in poorly textured image regions. The recent ACMMP approach enhances MVS depth estimation with multi-scale geometric consistency and a planar prior to reduce ambiguity in image regions with little texture information, resulting in denser depth maps.

The results achieved by FaSS-MVS with its three different SGM strategies, as well as the results obtained by the three other approaches, are listed in Table 5. While SGM^Π-pg^ outperforms the other two SGM extensions in terms of F-score, SGM^Π-sn^ has the lowest L1 error. This can be explained by the density of the depth maps. When SGM^Π-pg^ is used, more estimates pass the geometric consistency check, resulting in depth maps that are slightly more dense than those produced by SGM^Π^ and SGM^Π-sn^, increasing the F-score but also increasing the L1 error. Quantitatively speaking, however, the difference is only marginal, and a conclusion as to whether one particular SGM extension should be preferred over the others depends on the use case and should be drawn based on qualitative comparisons.

Compared to PSL in the configuration proposed by Häne et al. [31], i.e., without geometric post-processing, FaSS-MVS performs significantly better. This is due to the fact that the PSL does not include outlier removal, resulting in a high L1 error and a lower F-score. When combined with outlier filtering based on geometric consistency, the L1 of the resulting depth maps is the lowest of all the approaches evaluated, even lower than that of the two offline MVS approaches. However, the F-score is also significantly reduced due to the low completeness of the depth maps, as can be seen in Figure 10.

Offline MVS approaches are said to be superior to online approaches due to the availability of more input images and the absence of runtime constraints. And while COLMAP is slightly outperformed by PSL in combination with GCC in terms of the L1 error, it clearly achieves overall superiority in terms of the F-score and thus the trade-off between accuracy and completeness. When comparing the mean density of the resulting depth maps, ACMMP outperforms COLMAP by more than 24%. Surprisingly, however, ACMMP has a high L1 error and a low F-score. This suggests that the estimates computed by ACMMP in low-texture areas, where the other approaches do not provide estimates, are not very accurate. The significance of a comparison between online and offline MVS approaches can be questioned, however, since the two types of approaches make different assumptions and focus on different aspects within the processing, as further discussed in Section 4.1.

### 3.5. Runtime Comparison

As motivated above, the presented approach aims at incremental and online processing, i.e., the computation should ideally keep up with the input stream. Therefore, the total runtime of FaSS-MVS with its three SGM extensions compared to the comparable approach of PSL is evaluated in Table 6. In addition to the standard use of eight aggregation paths within the SGM optimization, which achieves the lowest error and has been used in previous experiments, the runtime and accuracy reduction of using only four aggregation paths is listed. This is motivated by a number of studies [35,36,60] that show that reducing the number of aggregation paths from eight to four can significantly reduce the computational time of SGM aggregation, while only marginally increasing its error. All measurements were conducted without any post-processing, i.e., DoG filtering or filtering based on geometric consistency.

The measurements clearly show that the PSL is much faster than FaSS-MVS. However, as the use-case-specific experiments in the following section show, the runtime of FaSS-MVS can vary greatly due to the variable number of sampling planes, depending on the distance between the input images and thus the observable depth range. The measurements also show that especially the SGM^Π-pg^ extension introduces a large computational complexity compared to SGM^Π^ and SGM^Π-sn^. However, reducing the number of aggregation paths has a large impact on the runtime, reducing it by up to 45%, while having only a marginal impact on the error. Whether the listed runtime is sufficient for online processing is further discussed in Section 4.3.

### 3.6. Use-Case-Specific Experiments Conducted on Real-World Datasets

Finally, to demonstrate the performance of the presented approach on use-case-specific and real-world datasets, experiments are performed using SGM^Π-pg^ with four aggregation paths and the above configuration on the TMB and FB datasets. Samples of the computed depth, normal, and confidence maps is shown in Figure 11, along with the corresponding depth maps estimated by COLMAP as a reference. The average processing time for the TMB dataset is 690 ms, but can vary between 320 ms and 1218 ms depending on the arrangement of the input data and the number of plane distances δ at which the scene is sampled. For the FB dataset, the average processing time is 800 ms, again varying between 514 ms and 1419 ms depending on the arrangement of the input images.

## 4. Discussion

In the following, the results of the conducted experiments are discussed with respect to different aspects, namely the overall accuracy (Section 4.1), the ability of the presented approach to reconstruct slanted surface structures (Section 4.2), the runtime and support for online processing (Section 4.3), and the effects of the post-filtering algorithms used and the relevance of the confidence estimates (Section 4.4).

### 4.1. Overall Accuracy

In Table 5, FaSS-MVS is evaluated against a comparable online MVS approach from the PSL. Considering the results achieved by the approach provided by the PSL, it is clearly outperformed by FaSS-MVS. The comparison is somewhat unfair, however, since the results presented by FaSS-MVS have undergone filtering based on a GCC. For this reason, additional experiments were performed in which the same filtering was applied to the PSL results, showing a superiority of the PSL with GCC over FaSS-MVS with respect to the L1 error. However, as suggested by the lower F-score and as shown by the extracts in Figure 10, the post-filtering leads to the removal of quite a few estimates in the depth maps computed by PSL. This again underlines the strength of FaSS-MVS in computing dense depth maps with high consistency and accuracy.

Furthermore, as the results in Table 5 show, the overall accuracies of the depth maps estimated by the presented approach are lower than those achieved by the two offline MVS approaches, i.e., COLMAP and ACMMP. This is not surprising, since the procedure and assumptions involved are very different between online approaches, such as the one presented in this paper, and offline approaches. Offline approaches assume that all input images are available at the time of reconstruction, allowing them to optimize the set of input images considered for the reconstruction of a given viewpoint. In contrast, online approaches, which perform MVS incrementally, only consider input images within a temporally limited window, at most all images acquired up to a certain point in time. In addition, offline approaches typically do not have time constraints either. Nevertheless, the quantitative differences between the results obtained with the presented approach and COLMAP are not that large, less than an order of magnitude, and even exceed those obtained with ACMMP. Furthermore, a qualitative comparison on use-case-specific input data makes the results of the presented approach very satisfactory. Compared to the geometric depth maps of COLMAP, the depth maps of SGM^Π-pg^ lack the fine-grained details, such as the roof structures in rows 5 and 6 of Figure 11, which are caused by the coarse-to-fine processing. However, larger structures are well represented and the quality of their reconstruction is comparable, as is the overall density. Although the fronto-parallel bias of SGM is reduced, some artifacts of fronto-parallel sampling are still visible, especially in the normal maps of Figure 11.

### 4.2. Ability to Account for Non-Fronto-Parallel Surfaces

To further increase the accuracy of the reconstruction of slanted, non-fronto-parallel surface structures, this work proposes, in addition to SGM^Π^, two extensions to the SGM algorithm that should reduce the fronto-parallel bias. Namely, the incorporation of surface normals to adjust the zero-cost transition in the SGM path aggregation (SGM^Π-sn^) and the penalization of deviations from the gradient of the minimum-cost path (SGM^Π-pg^). The experiments conducted show that these extensions provide only a slight quantitative improvement over the standard SGM adaptation (SGM^Π^) to plane-sweep sampling. This finding is in contrast to the experiments of Scharstein et al. [41]. There are at least two reasons for this discrepancy. First, Scharstein et al. [41] demonstrate their implementation on a two-view stereo dataset, where the input images are captured by two cameras mounted on a fixed rig and oriented in the same direction. In addition, prior to processing, the images are rectified, i.e., transformed, so that both lie in the same image plane and the epipolar lines coincide with the image rows. Thus, in the dense image matching process, the images are sampled equidistantly with a step size of 1 pixel. In the case of the presented approach, however, the distances of the sampling planes and thus the sampling points are chosen in such a way that the disparity shift along the epipolar line between two consecutive planes is less than or equal to 1. This results in sampling with a much higher density, which already reduces the staircase effect in the case of SGM^Π^. And secondly, Scharstein et al. [41] proposes to use a ground truth normal map to adjust the zero-cost transition, whereas in the presented approach, the upscaled normal map of the previous iteration of the hierarchical processing is used. This is bootstrapped with SGM^Π^ at the highest pyramid level, which introduces inaccuracies that probably cannot be fully compensated for. However, the qualitative analysis shows that SGM^Π-sn^ and SGM^Π-pg^ clearly lead to smoother normal maps and reduce staircase artifacts in the depth maps, which is why only SGM^Π-pg^ is considered in the use-case-specific experiments.

In addition to reducing the fronto-parallel bias in the SGM path aggregation, the plane-sweep algorithm in our approach allows us to adapt the image matching to the surface structures in the scene by selecting appropriate normal vectors and sweeping directions. In a short qualitative experiment (see Figure A1), the effects of a horizontal plane sampling compared to a fronto-parallel sampling, both in combination with SGM^Π^, are investigated. The results show that horizontal sampling leads to more consistent depth estimates with little or no staircasing artifacts in areas where the surface structure coincides with the plane orientation, e.g., the ground plane. However, in areas where the surface structure is not horizontal, non-fronto-parallel sampling introduces significant errors. To overcome this effect, one can consider dividing the scene into local regions that are individually sampled with different plane orientations, similar to the local-plane-sweep approach presented by Sinha et al. [38]. However, this comes at the cost of higher computational complexity. Another remedy is to repeat the plane-sweep image matching several times on the whole image domain prior to the SGM optimization, with different sweeping directions, and to perform a pixel-wise pre-selection of the best plane orientation based on the matching costs, similar to the approach of Pollefeys et al. [25]. This results in a smaller increase in computational complexity compared to the first option.

### 4.3. Runtime and Online Processing

Given the runtime measurements in Table 6, the presented approach is obviously not capable of real-time and low-latency processing, in the sense that for each input frame, a depth map is computed at similar frame rates as given for the input stream. However, considering the nature of the approach and the expected input data, the runtime is generally sufficient for online processing, which will be explained in the following section. The presented approach takes a bundle of three or more input images, with a bundle size of five images actually yielding better results, and performs MVS on a reference image of the input bundle, typically the middle one. While these input images could be provided by individual cameras, it is assumed that the images are extracted from an input sequence captured by a single camera moving around a static scene. In addition, not every frame of the input sequence can be used, since a suitable baseline must lie between each input frame to enable scene depth estimation. This, of course, depends on the depth range to be sampled and the scene structure. In the case of the TMB dataset, the average distance between the individual input images is 1.8 m and 1.03 m for a flight altitude of 15 m and 8 m, respectively. This increases at higher altitudes due to greater scene depth. Modern COTS rotor-based UAVs can fly up to a speed of over 10 m/s. However, the typical flight speed for image acquisition is closer to 1–3 m/s [61,62]. Thus, if the sets of input images are disjoint, then an estimation needs to be performed at least every 3 s, considering a low flight altitude together with a high flight speed of about 3 m/s and an input bundle size of three images. If a maximum overlap between the input bundles is desired, i.e., a new depth map estimation is triggered with each new suitable input frame and it reuses four images from the previous bundle, the required runtime is significantly lower. However, as the use-case-specific experiments for the TMB and FB datasets show, the average processing time of SGM^Π-pg^, which is the most computationally expensive variant, is between 1 and 2 Hz, depending on the arrangement of the input images. Another way to reduce the runtime is to use higher Gaussian pyramids, which again comes at the cost of a reduced level of detail, as already pointed out in the discussion on overall accuracy (see Section 4.1). In short, there are a number of possible settings both in the acquisition of the input data, e.g., regarding the flight speed or the size and overlap of the input bundles, and in the configuration of the presented approach, e.g., regarding the Gaussian pyramid height, the depth range or the optimization strategy, which allow us to adapt the runtime to the rate of the input images and thus to allow online processing.

The emergence of high-performance systems-on-a-chip (SoCs) with embedded GPUs, such as the NVIDIA Jetson series, allows approaches like FaSS-MVS to be brought directly to the sensor platform, e.g., the UAV, for on-board processing. To evaluate the feasibility of running FaSS-MVS on-board an embedded device, some additional runtime measurements were performed on the NVIDIA Jetson AGX, equipped with an 8-core 64-bit ARMv8.2 CPU and a 512-core Volta GPU. On an excerpt of the TMB dataset with an image size of 1920×1080 pixels, FaSS-MVS with SGM^Π^ achieves an average runtime of 727 ms on the Jetson AGX, compared to an average runtime of about 403 ms on the NVIDIA Titan X. As already discussed, the runtime can be further reduced by increasing the pyramid height to n=4 and n=5, for example, while accepting a decrease in the quality of the results. This results in average runtimes of 444 ms and 385 ms, respectively. These experiments show that FaSS-MVS is capable of on-board processing using a high-performance embedded SoC such as the NVIDIA Jetson AGX. This may be of particular interest when considering deployment on a sensor carrier that does not suffer from severe power constraints.

### 4.4. Post-Filtering and the Relevance of the Estimated Confidence Values

A comparison of the L1 errors in Table 5 with those listed in Table 2 shows that using post-filtering based on geometric consistency can drastically reduce the mean errors by about 40%. The trade-off for this improvement is a loss of density in the depth map and an increase in latency between input and results. The latter is due to the additional sliding window introduced by geometric consistency-based filtering. In addition to the bundle of input images for which only one set of estimates is produced, the geometric filter also requires two or more depth maps for processing. The geometric filter is also more computationally expensive than the DoG filter. Again, whether to use the DoG or the geometric filter depends on the application. For example, if the presented approach is used for the task of online 3D reconstruction, i.e., a subsequent depth map fusion step is used [7], the geometric-consistency-based filtering is typically performed in the depth map fusion and can thus be omitted. The DoG filter, on the other hand, is very efficient and does not introduce any additional latency. However, as mentioned in the experiments, the DoG filter may also remove potentially good estimates, since it is performed only on the data provided by the input image. Nevertheless, the DoG filter is of great benefit, especially when working with input data containing many homogeneous areas with little or no texture, e.g., a clear or cloudy sky in case of extreme viewing angles.

Finally, as a third output, the presented approach computes a confidence map containing pixel-wise confidence values corresponding to the depth estimates. In this work, these confidence measures are used to perform a comparison between the different SGM extensions based on an ROC analysis (see Figure 8). As noted above, the fact that some of the curves are not monotonically increasing suggests that the confidence values do not adequately represent the certainty of the estimates. For example, the fact that the scene in row 6 of Figure 11 consists mostly of fronto-parallel structures leads to a confidence map with high certainty values, while the confidence map in row 2 of Figure 11 makes the estimation of the roof of the building, which appears qualitatively very accurate, completely uncertain. A similar observation can be seen in the confidence maps shown in Figure 6. Only because the ground plane is highly tilted with respect to the image plane, the confidence of the corresponding estimates becomes very low, even though they do not appear qualitatively more accurate than the estimates on the building facade. The most likely reason is that modeling a confidence score based on surface orientation alone is not very meaningful. Incorporating additional heuristics based on internal properties of the algorithm, as carried out in previous work [12], could improve the confidence estimation, but this still requires a cumbersome empirical study of the hyper-parameters. In recent years, however, the performance of learning-based approaches to confidence estimation [63,64] has improved significantly. They are often agnostic to the internals of the algorithm and can be trained on any data for which both estimated and reference depth maps are available.

## 5. Conclusions

In conclusion, we present an approach for multi-view stereo (MVS) from UAV-borne imagery that allows for fast, dense, and incremental 3D mapping. This approach consists of a hierarchical processing scheme that estimates dense depth maps and corresponding normal and confidence maps. For the depth map computation, dense multi-image matching using the plane-sweep algorithm is used to generate pixel-wise depth hypotheses. From these hypotheses, a dense depth map is extracted using the optimization scheme of the widely used Semi-Global Matching (SGM) algorithm. Here, the SGM algorithm is not only adapted to work with the multi-image matching of the plane-sweep algorithm, but also extended to reduce the fronto-parallel bias and to account for slanted surface structures by introducing two additional regularization schemes. The successive normal and confidence map estimation is performed separately on the results of the depth estimation. In a final filtering step, geometric consistency is enforced over multiple depth maps, which greatly increases the overall accuracy of the resulting depth maps.

The performance of our approach is quantitatively evaluated on two public datasets containing image data of model-scaled scenes captured from an aerial perspective and providing accurate ground truth. The experiments show that for the best configuration, the estimated depth maps have a mean absolute L1 error of only 8.5 mm on the DTU dataset, or 1%, with respect to the maximum depth of the reconstructed scene. In comparison, on the same dataset, the geometric depth maps from COLMAP, a widely used open-source toolbox for offline MVS, have a mean absolute error of 3.8 mm. Thus, even though the presented approach does not have all image data of the input sequence available at the time of reconstruction and is subject to runtime constraints to ensure fast and online processing, its quantitative results are not too far off from state-of-the-art offline approaches. While the quantitative results do not show a significant improvement by the presented SGM extensions to account for slanted surface structures, a qualitative comparison reveals their ability to account for non-fronto-parallel surfaces. Thus, in the case of oblique aerial imagery containing many slanted surfaces, the presented SGM extension, which penalizes deviations from the gradient of the minimum-cost path, i.e., SGM^Π-pg^, is the best choice, despite its higher computational complexity. Final experiments on real-world and use-case-specific datasets have shown that the presented approach is well suited for online processing in terms of runtime, achieving a processing rate of 1–2 Hz, meaning that it keeps up with the monocular input stream and allows for incremental 3D mapping as input data are received. Fast 3D mapping, in turn, can facilitate other important applications or tasks, such as rapid assessment of inaccessible areas by emergency responders, e.g., after a flood or earthquake, to perform disaster relief or search and rescue missions.

Finally, there are also some aspects to consider for future work. Although the approach supports different plane orientations in plane-sweep multi-image matching, each estimation is performed with only one orientation. In the future, the approach should be extended to use multiple plane orientations within the computation of a single depth map. This will allow for smoother reconstruction of large planar surfaces such as the ground plane, but will also allow for higher accuracy in other regions by using fronto-parallel sampling. Furthermore, while we note that the processing speed is sufficient, a further reduction in runtime and a more efficient use of GPU resources would free up more opportunities for other concurrent tasks, such as depth map fusion or orthophoto generation. Therefore, further optimization in terms of runtime and utilization of processing resources is an ongoing task. In addition, due to the ongoing development and rapid advancement of deep learning-based approaches for the task of MVS, we want to investigate whether individual steps or even the entire approach can be replaced by an appropriate learning-based approach, while maintaining the reliability for use in the context of critical applications. Finally, the work of Nex and Rinaudo [65] has shown that the complementary use of Light Detection and Ranging (LiDAR) and image-based techniques for photogrammetric tasks has great potential. In addition, with the improvements in LiDAR sensors and the ability to equip commercial UAVs with such sensors, such as the Zenmuse L1, their use to facilitate fast and incremental 3D mapping will inevitably be considered in future work.

## Figures and Tables

**Figure 1 sensors-24-06397-f001:**
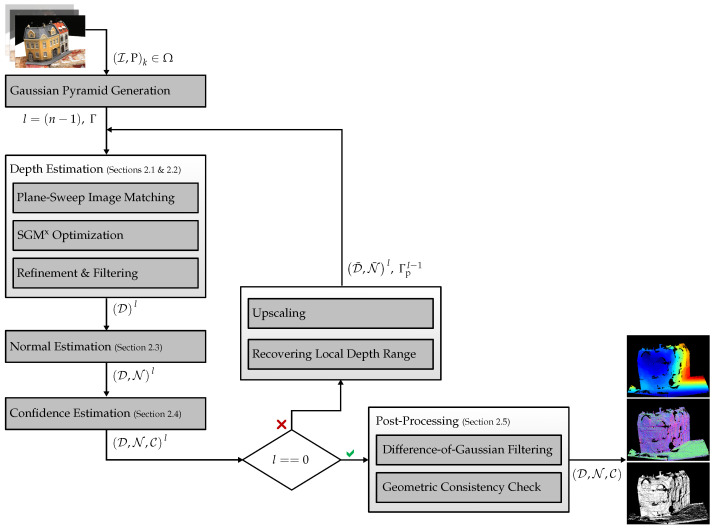
Overview of the processing pipeline for FaSS-MVS. Given a bundle of images and corresponding camera poses ${\left(I,P\right)}_{k}$ of an input sequence, a hierarchical MVS estimation is performed to recover a depth, normal and confidence map $\left(D,N,C\right)$. Adapted from [12,45].

**Figure 2 sensors-24-06397-f002:**
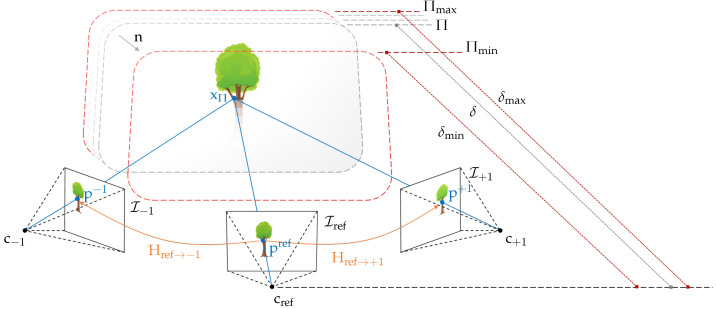
Illustration of the plane-sweep algorithm for multi-image matching. A scene is sampled by a plane Π=(n,δ), where n is the normal vector of the plane and δ is the orthogonal distance of the plane from $c_{\mathrm{ref}}$. The plane is swept through space along its normal vector between two bounding planes $\Pi_{\max}$ and $\Pi_{\min}$. For each distance δ of Π, the reference pixel $p^{\mathrm{ref}}$ is projected by the plane-induced homography $H_{\mathrm{ref}\to k}$ into an arbitrary number of viewpoints where it is matched with the corresponding pixel in $I_{k}$.

**Figure 3 sensors-24-06397-f003:**
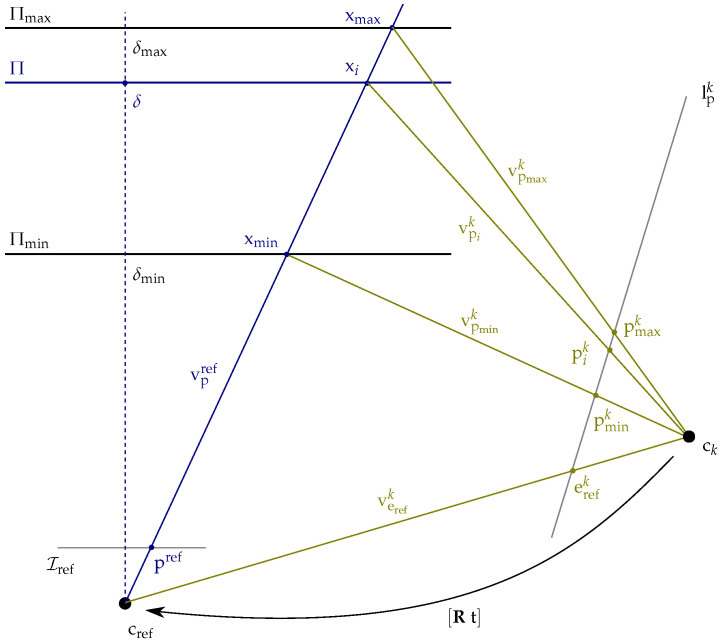
Illustration of determining the orthogonal distance parameter of the sampling planes of the plane-sweep multi-image matching by using the cross-ratio and epipolar geometry. Here, $c_{\mathrm{ref}}$ and $c_{k}$ represent the positions of the optical centers of the two cameras. Adapted from [11].

**Figure 4 sensors-24-06397-f004:**
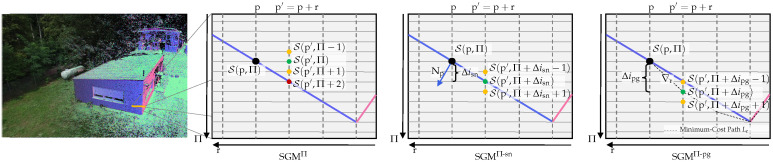
Illustration of the different path aggregation strategies along one path direction r within the three presented SGM^x^ optimization schemes. Column 1: Reference image and normal map of a building. Illustrated area is marked with yellow line. Column 2: SGM^Π^ path aggregation. The blue and pink lines represent the blue and pink surface orientations on the building facade. When aggregating the path costs for pixel p at plane Π, SGM^Π^ will include the previous costs at the same plane position (green) without additional penalty. The previous path costs at Π±1 (yellow) will be penalized with $\varphi_{1}$. The previous path costs located at Π+2 (red), which is actually located on the corresponding surface, will be penalized with the highest penalty $\varphi_{2}$. Column 3: SGM^Π-sn^ uses the normal vector $n_{p}$, which encodes the surface orientation at pixel p, and computes a discrete index jump $\Delta i_{\mathrm{sn}}$, which ideally adjusts the zero-cost transition so that the previous path costs at Π+2 are not penalized. Column 4: Similar to SGM^Π-sn^, SGM^Π-pg^ adjusts the zero-cost transition. However, the discrete index jump $\Delta i_{\mathrm{pg}}$ is derived from the running gradient ∇r of the minimum-cost path. Adapted from [12].

**Figure 5 sensors-24-06397-f005:**
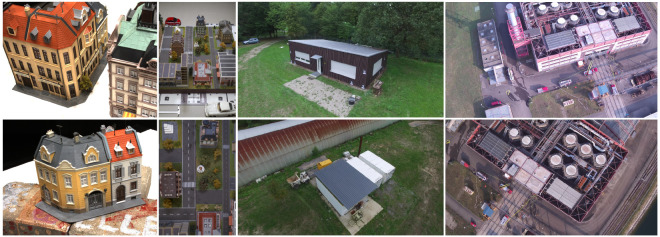
Overview of the datasets used for performance evaluation of FaSS-MVS. Column 1: Two building models from the DTU Robot MVS dataset. Column 2: Example images in oblique and nadir view from the 3DOMcity Benchmark dataset. Column 3: Excerpt of the privately acquired TMB dataset. Column 4: Use-case-specific dataset acquired during an exercise of the local fire brigade.

**Figure 6 sensors-24-06397-f006:**
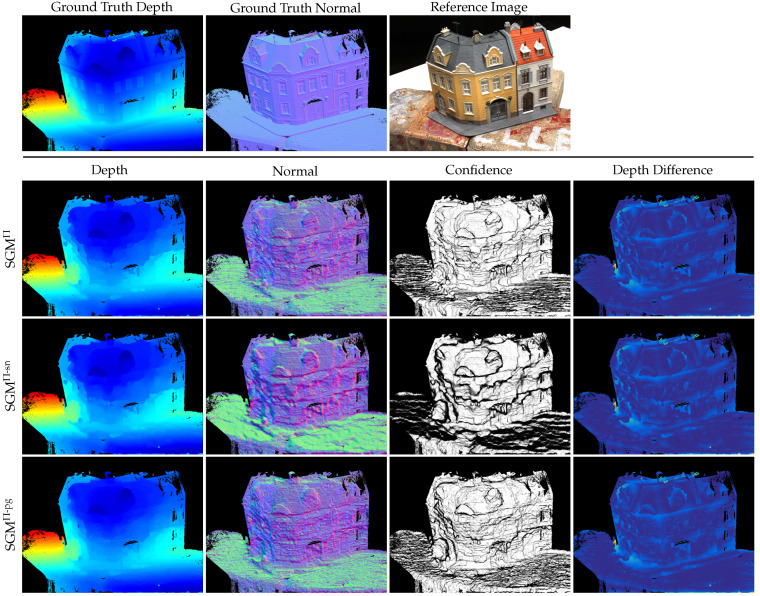
Qualitative comparison of the results achieved by the three different SGM implementations on the DTU dataset. Row 1: Reference data from the dataset, i.e., the ground truth depth and normal map, as well as the reference image for which the data are computed. Rows 2–4: Data, i.e., depth, normal and confidence maps, computed by SGM^Π^, SGM^Π-sn^ and SGM^Π-pg^, respectively. Furthermore, difference maps are provided which hold the pixel-wise absolute difference between the estimated depth map and the ground truth. The color encoding reaches from dark blue (low error) via green to yellow (high error). The depth range within the depth maps reaches from 580 mm (blue) to 830 mm (red). The estimated maps are masked according to the ground truth.

**Figure 7 sensors-24-06397-f007:**
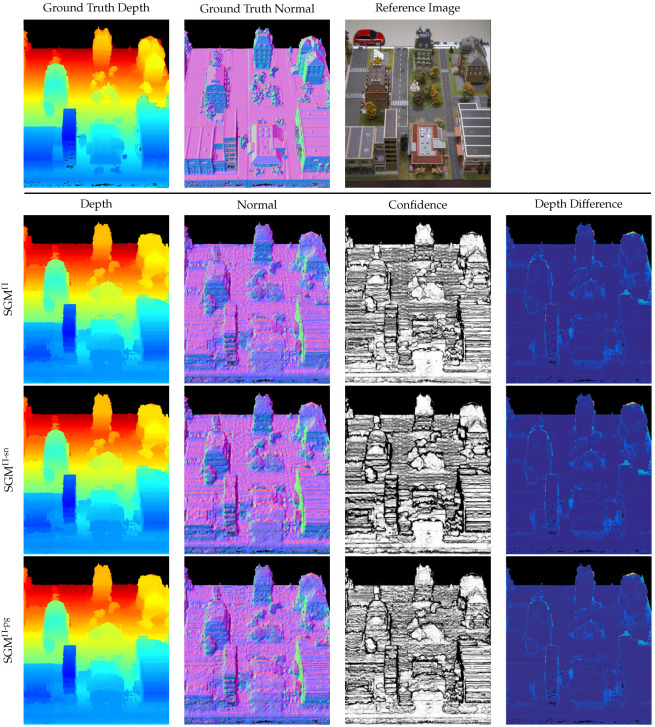
Qualitative comparison of the results achieved by the three different SGM implementations on the 3DOMcity dataset. Row 1: Reference data from the dataset, i.e., the ground truth depth and normal map, as well as the reference image for which the data are computed. Rows 2–4: Data, i.e., depth, normal and confidence maps, computed by SGM^Π^, SGM^Π-sn^ and SGM^Π-pg^, respectively. Furthermore, difference maps are provided which hold the pixel-wise absolute difference between the estimated depth map and the ground truth. The depth range within the depth maps reaches from 1 m (blue) to 1.8 m (red). The estimated maps are masked according to the ground truth. For visualization in this figure, the resulting images have been rotated counterclockwise by $90^{\circ }$. Thus, the color encoding of the normal maps differs from that used in the other figures. Here, red represents an upwards orientation, while green represents an orientation to the left.

**Figure 8 sensors-24-06397-f008:**
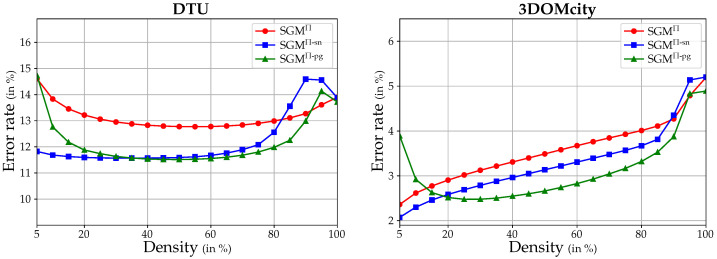
ROC curves illustrating the error rate achieved by the three different SGM implementations as a function of increasing density of the estimated depth map.

**Figure 9 sensors-24-06397-f009:**
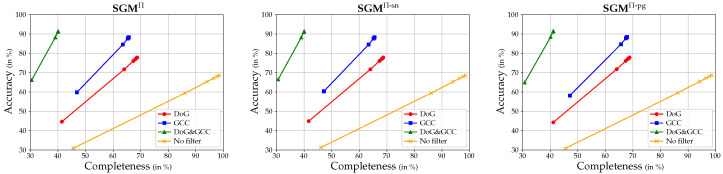
Accuracy–completeness curves of different post-filtering strategies, i.e., DoG filtering, GCC as well as a combination of both, executed in combination with the three different SGM extensions and a fronto-parallel sampling. In this, the threshold θ is varied within the list of {1.25,1.20,1.15,1.10,1.05,1.01}. By decreasing θ, the accuracy and completeness rates drop.

**Figure 10 sensors-24-06397-f010:**
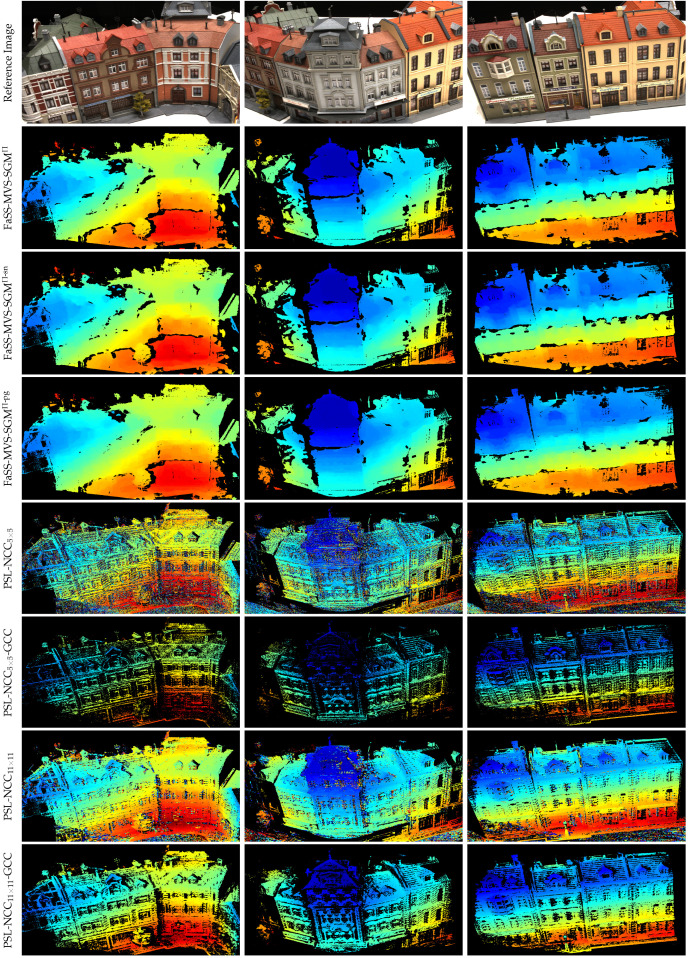
Qualitative comparison of FaSS-MVS with its three SGM extensions and GCC, the PSL with differently sized support regions for the NCC, as well as with GCC.

**Figure 11 sensors-24-06397-f011:**
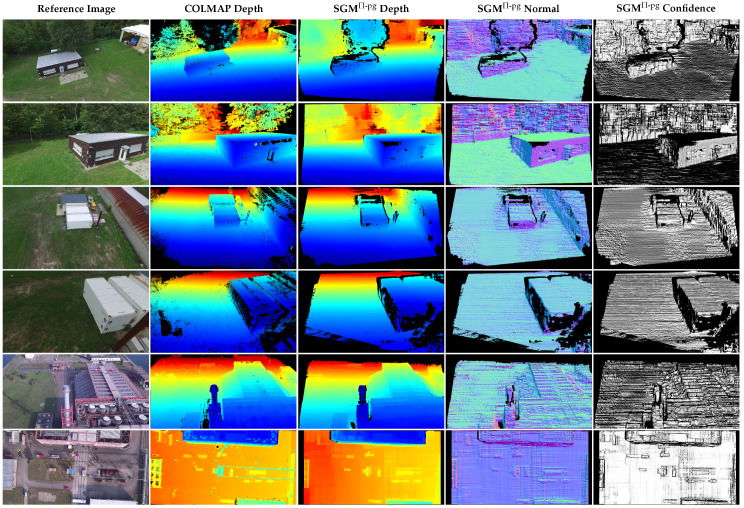
Qualitative results of SGM^Π-pg^ with 4 aggregation paths achieved on the two real-world and use-case-specific datasets, namely the TMB dataset and the FB dataset. As comparison, the corresponding depth maps estimated by COLMAP are also visualized. Rows 1 and 2: TMB Building scene captured from an altitude of 15 m and 8 m, respectively. Rows 3 and 4: TMB Container scene. Rows 5 and 6: Two excerpts from the FB dataset.

**Table 1 sensors-24-06397-t001:** Summary of the key characteristics of the four evaluation datasets. *(i): intrinsic calibration, (e): extrinsic calibration.*

Dataset	# Images	Image Size	Calibration	Reference	Scene	Flight Pattern
DTU	1029	1600 × 1200	(i) pre-calibration, (e) pre-calibration	structured-light sensor	scale-modeled buildings	orbital
3DOMcity	245	1798 × 1200	(i) COLMAP, (e) pre-calibration	semi-global offline DIM	scale-modeled urban area	linear
TMB	2013	1920 × 1080	(i) COLMAP, (e) COLMAP	COLMAP (geometric depth)	rural area	orbital
FB	202	1920 × 1080	(i) COLMAP, (e) COLMAP	COLMAP (geometric depth)	industrial area	linear

**Table 2 sensors-24-06397-t002:** Quantitative comparison of the results obtained by different implementations and adaptations of the SGM algorithm in combination with a fronto-parallel sweeping direction. The best results are underlined.

Dataset	Metric	SGM^Π^	SGM^Π-sn^	SGM^Π-pg^
DTU	L1-abs	19.832	19.768	19.684
	(in mm)	±16.225	±16.192	±16.154
	L1-rel	0.027	0.027	0.027
		±0.021	±0.021	±0.021
3DOMcity	L1-abs	14.615	14.673	15.074
	(in mm)	±6.254	±6.229	±6.133
	L1-rel	0.012	0.012	0.012
		±0.007	±0.007	±0.006

**Table 3 sensors-24-06397-t003:** The AUC together with the AUC-Opt. and the difference between those two (ΔAUC) for the three different SGM implementations. The best results are underlined.

Dataset	Metric	SGM^Π^	SGM^Π-sn^	SGM^Π-pg^
DTU	AUC	1245.5	1157.5	1150.3
	AUC-Opt.	180.5	180.5	180.8
	ΔAUC	1065.0	977.0	969.6
3DOMcity	AUC	338.0	312.2	290.0
	AUC-Opt.	192.6	192.6	193.0
	ΔAUC	145.4	119.6	96.9

**Table 4 sensors-24-06397-t004:** F-scores achieved by the SGM^x^ approaches together with the post-filtering based on GCC. The best results are underlined.

Approach	$F_{1.25}$	$F_{1.20}$	$F_{1.15}$	$F_{1.10}$	$F_{1.05}$	$F_{1.01}$
	(in %)	(in %)	(in %)	(in %)	(in %)	(in %)
SGM^Π^	74.2	74.1	74.0	73.7	71.5	51.9
SGM^Π-sn^	74.1	74.1	74.0	73.6	71.5	52.3
SGM^Π-pg^	75.6	75.5	75.4	75.1	72.9	51.4

**Table 5 sensors-24-06397-t005:** Quantitative comparison of FaSS-MVS with its three SGM extensions, combined with post-filtering based on geometric consistency checking, with related approaches from the literature on the data of the DTU benchmark. As a reference, the results of the PSL [31] for online MVS, with differently sized support regions for the NCC, as well as with GCC are given. The results of two offline MVS approaches, namely COLMAP [4] and ACMMP [18], are provided for reference. The best results are underlined.

Approach	L1-abs	L1-rel	$F_{1.25}$	$F_{1.20}$	$F_{1.15}$	$F_{1.10}$	$F_{1.05}$	$F_{1.01}$
	(in mm)		(in %)	(in %)	(in %)	(in %)	(in %)	(in %)
FaSS-MVS-SGM^Π^	8.549±7.509	0.012±0.011	74.2	74.1	74.0	73.7	71.5	51.9
FaSS-MVS-SGM^Π-sn^	8.479±7.559	0.012±0.011	74.1	74.1	74.0	73.6	71.5	52.3
FaSS-MVS-SGM^Π-pg^	8.722±7.255	0.013±0.010	75.6	75.5	75.4	75.1	72.9	51.4
${PSL\text{-}NCC}_{5\times 5}$	73.924±23.686	0.106±0.027	67.3	62.5	56.5	48.4	35.3	9.9
${PSL\text{-}NCC}_{5\times 5}\text{-}GCC$	2.32±1.26	0.003±0.002	32.5	32.5	32.5	32.5	32.4	31.2
${PSL\text{-}NCC}_{11\times 11}$	51.229±28.209	0.071±0.035	72.2	69.5	66.0	61.1	51.0	21.9
${PSL\text{-}NCC}_{11\times 11}\text{-}GCC$	2.17 ± 1.23	0.003 ± 0.002	61.1	61.1	61.1	61.1	61.0	58.9
COLMAP	3.745±5.498	0.006±0.004	80.2	80.2	80.1	80.0	79.6	74.4
ACMMP	12.963±13.379	0.018±0.018	77.5	77.2	76.6	75.5	73.0	55.2

**Table 6 sensors-24-06397-t006:** Runtime comparison of FaSS-MVS with its three SGM extensions and the comparable approach from the PSL [31]. Measurements were performed on the DTU benchmark dataset and represent the average runtime required by the different approaches to estimate a single depth map. With respect to FaSS-MVS, the difference between using 8 and 4 aggregation paths within the SGM optimization is also evaluated. The best results are underlined.

Metric	FaSS-MVS-SGM^Π^		FaSS-MVS-SGM^Π-sn^		FaSS-MVS-SGM^Π-pg^		PSL
	8 **-Path**	4 **-Path**	8 **-Path**	4 **-Path**	8 **-Path**	4 **-Path**	${\mathrm{NCC}}_{5\times 5}$
Runtime (in ms)	640	413	895	546	2079	1132	139
Δ L1-abs (in %)		+6.3		+6.7		+6.2	

## Data Availability

Publicly available datasets were analyzed in this study. This data can be found here: https://roboimagedata.compute.dtu.dk/?page_id=36 (accessed on 27 August 2024) and https://3dom.fbk.eu/3domcity-benchmark (accessed on 27 August 2024).

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
