# Peer review of "FaSS-MVS: Fast Multi-View Stereo with Surface-Aware Semi-Global Matching from UAV-Borne Monocular Imagery"

_sensors, 2024, doi:10.3390/s24196397_

Round 1

Reviewer 1 Report

Comments and Suggestions for Authors

Dear Authors,

the manuscript describes image-matching methods extensively, extending with your know-how. In this context, I consider the work beneficial and partially broadening the field of knowledge in the area. I agree with the methods used in evaluation e.g. completeness and accuracy of compared depth maps and consider them to be sufficiently robust and supporting the findings summarized in the conclusions.  Although the authors set as one of their objectives to give a detailed description of the algorithms used, from my point of view it is not necessary to describe well-known algorithms like semi-global matching. The authors should focus only on the novelty of the study that it brings, so there is a possibility to reduce the length of the manuscript.  The description of the datasets and methods for statistical evaluation should be the part of the material and methods chapter. I consider the conclusions and discussion to be sufficient, supported with specific recommendations, e.g. for what surfaces is this method suitable.

I give the editors and authors to consider whether it would be appropriate to shorten the manuscript and the stated objectives, since as I have already mentioned the manuscript is quite extensive, based on the author's dissertation, and thus loses a bit of the readability and character of a research article.

Author Response

Comments 1: The authors should focus only on the novelty of the study that it brings, so there is a possibility to reduce the length of the manuscript.
Response 1: Thank you for bringing this to our attention. We have once again critically reviewed the content of the article and reduced its length by 5 pages.

Comments 2: The description of the datasets and methods for statistical evaluation should be the part of the material and methods chapter.
Response 2: Thank you for your comments. We have moved the sections on evaluation datasets and evaluation metrics to the 'Materials and Methods' section.

Reviewer 2 Report

Comments and Suggestions for Authors

1.Currently, the number of pixels in the onboard cameras of UAV is generally large, and the number of pixels used in the experimental analysis in the manuscript is generally low. Can you increase the experimental data by more than 40 million pixels? Will this have a serious impact on computational efficiency.

2. The algorithm in the article basically did not use deep learning methods. The author spent a large amount of space discussing the deep learning part, which has a weak correlation with this article. Can you reduce the weight of this aspect.

3. An important objective of this research is to provide real-time 3D scene generation for firefighting operations. However, during firefighting operations, the scene is usually relatively complex, such as fires, which may have interference factors such as smoke and fire. Can you increase experimental data that is more relevant to firefighting rescue scenes.

4. A small question, in the conclusion section, the author uses the term "absolute accuracy", but the generated accuracy is strictly speaking "relative accuracy" or directly expressed as "accuracy".

Author Response

Comments 1: Can you increase the experimental data by more than 40 million pixels? Will this have a serious impact on computational efficiency.
Response 1: Thank you for your comments. Unfortunately, due to the limited time for a minor revision, we are not able to increase the experimental data and run more experiments. However, increasing the image size will intuitively have a significant impact on computation time, as more depth estimates will need to be computed.

Comments 2: The author spent a large amount of space discussing the deep learning part, which has a weak correlation with this article. Can you reduce the weight of this aspect.
Response 2: Thanks for pointing this out. We have shortened the section on related work in deep learning. We also critically reviewed the content of the article and reduced its length by 5 pages.

Comments 3: Can you increase experimental data that is more relevant to firefighting rescue scenes.
Response 3: Thank you for your ideas. As mentioned above, time is too short to do further experiments. We are aware that the scenes we have been working on are not complex enough, but unfortunately we have so far not been able to capture more relevant data from firefighting operations.

Comments 4: A small question, in the conclusion section, the author uses the term "absolute accuracy", but the generated accuracy is strictly speaking "relative accuracy" or directly expressed as "accuracy".
Response 4: Thank you for your question. During our experiments we calculated a number of error measures. One of them is an absolute L1 error, which we refer to in our conclusion. Yes, you are right, the accuracy we calculated is a relative measure. But we believe you are referring to the part where we state that our approach achieves an "absolute L1 error of only 8.5 mm".

Reviewer 3 Report

Comments and Suggestions for Authors

This work presents a fast, surface-aware semi-global optimization approach for multi-view stereo(FaSS-MVS) that enables for rapid depth and normal map estimation from monocular aerial video data captured by unmanned aerial vehicles(UAVs).

The logic of the manuscript is clear, the implementation and methodology of the proposed approach is elaborated in detail, and the discussion is profound and specific.

No errors and issues needing revision were found.

In summary, this work has high academic value and the paper is well written. Suggest accepting in its current form.

Author Response

Thank you for your review.